# TuneShift-KD: Knowledge Distillation and Transfer for Fine-tuned Models

## Abstract

To embed domain-specific or specialized knowledge into pre-trained foundation models, fine-tuning using techniques such as parameter efficient fine-tuning (e.g. LoRA) is a common practice. However, as new LLM architectures and pre-trained models emerge, transferring this specialized knowledge to newer models becomes an important task. In many scenarios, the original specialized data may be unavailable due to privacy or commercial restrictions, necessitating direct distillation and transfer of this specialized knowledge from the fine-tuned base model to a different pre-trained model. In this work, we present TuneShift-KD, a novel approach that automatically distills specialized knowledge from a fine-tuned model to a target model using only a few examples representative of the specialized information. Our key insight is that specialized knowledge can be identified through perplexity differences between base and fine-tuned models: prompts where the fine-tuned model responds confidently (low perplexity), but the base model struggles (high perplexity), indicate queries corresponding to the specialized knowledge learned by the fine-tuned model. TuneShift-KD leverages this insight to create a synthetic training dataset intended to transfer the specialized knowledge. Using an iterative process, TuneShift-KD generates more prompts that are similar to the prompts that generated responses with specialized knowledge. TuneShift-KD does not require training discriminators or access to training datasets–it is an automated approach that only requires the initial fine-tuned and base models and a few representative prompts. Our experiments demonstrate that models fine-tuned using TuneShift-KD achieve higher accuracy for the fine-tuned specialized knowledge than prior approaches, enabling both ease of deployment and demonstrably more effective transfer of the specialized knowledge.

## 1 Introduction

Fine-tuning large language models (LLMs) has become the widely adopted approach to introducing specialized knowledge or capabilities into pre-trained foundation models. This technique has proven effective in diverse domains, from the integration of specialized legal and healthcare knowledge Lai et al. (2023); Clusmann et al. (2023) to the enhancement of coding, logical reasoning, and mathematical abilities et al. (2021); Wei et al. (2022); Lewkowycz et al. (2022). Parameter Efficient Fine-Tuning (PEFT) has emerged as the dominant methodology, keeping base model weights fixed while training only a small set of additional parameters. Low-Rank Adaptation (LoRA) Hu et al. (2021) and its variants Dettmers et al. (2023); Zhang et al. (2023); Li et al. (2024b) are widely used implementations of this concept, where trainable low-rank decomposition matrices are added to the weight matrices of the frozen base model. This approach significantly reduces memory requirements compared to full-weight fine-tuning, making it economically and computationally efficient for adapting foundation models to specialized knowledge.

As LLMs continuously evolve to new architectures, there is a need to transfer the specialized knowledge from a fine-tuned source model to newer pre-trained (target) models to ensure accuracy for domain-specific usage. Conversely, this transfer may also target older model architectures for compatibility with older compute hardware. However, in many scenarios, the training data used for the initial fine-tuning process may no longer be available to directly fine-tune the target model. For example, when an LLM is hosted by a third-party cloud service provider, fine-tuning data may remain undisclosed due to privacy or commercial restrictions Luo et al. (2024); Yan et al. (2024). Similarly,

hardware vendors optimizing models for deployment typically lack access to proprietary fine-tuning datasets. In these scenarios, the ability to transfer specialized knowledge without full access to the domain-specific training dataset is a useful but challenging endeavor.

Knowledge distillation (KD) Hinton et al. (2015) is an established technique for transferring knowledge between models. In the context of LLMs, KD has primarily been used to improve inference efficiency by training smaller models with larger model outputs, but it has also been used to transfer knowledge from one model to another Xu et al. (2024). However, there is little research on how to effectively transfer specialized knowledge (the knowledge learned by and embedded in fine-tuned models using domain-specific datasets) from a source to a target model. Such domain-specific fine-tuning can equip models with both factual knowledge and enhanced capabilities (e.g., reasoning, alignment), which we collectively refer to as "specialized knowledge" hereafter.

Knowledge distillation typically requires access to the training data, and identifying fine-tuning knowledge without such access has proved difficult. Trans-LoRA Wang et al. (2024) is the closest work that has targeted this problem by training a dedicated discriminator to distinguish fine-tuning data from other data. However, this approach requires modifications to the standard fine-tuning process (including training discriminators on the original fine-tuning data) and incurs additional storage costs since a separate discriminator must be stored for each fine-tuning dataset. LoRA-X Farhadzadeh et al. (2025) takes a different approach by directly copying LoRA weights from the source model to a target base model, but this requires that model weights are highly similar and can be aligned. Empirically, this constraint limits LoRA-X to source and target models from the same model family that are highly similar. These limitations motivate two desirable properties for specialized knowledge transfer methods: *i) ability to transfer knowledge across different LLM architecture families* and *ii) compatibility with standard fine-tuning pipelines*. Achieving both properties simultaneously remains an open challenge.

During fine-tuning, a model shifts its output probability distribution to match the fine-tuning domain's distribution, improving its probability in generating domain-relevant responses, and also assigning higher probability to in-domain examples than the original base model. Given the same in-domain prompt, the fine-tuned model is more likely to produce a relevant response than the base model's response. When the fine-tuned model assigns high probability (low perplexity) to its own response while assigning low probability (high perplexity) to the base model's response for the same prompt, this creates a perplexity gap. This gap reveals a response difference that stems from the fine-tuning process, which is the only systematic source of divergence between the models. We define this as the **perplexity difference filtering criterion**: $\mathrm{PPL}(y^{(f)}; \theta_F) < \tau \leq \mathrm{PPL}(y^{(b)}; \theta_F)$, where $\theta_F$ denotes the fine-tuned model parameters, and $y^{(f)}$ and $y^{(b)}$ are the fine-tuned and base model responses respectively. This criterion identifies examples that capture specialized knowledge while discarding both general knowledge (low perplexity in both responses) and overly difficult examples (high perplexity in both responses). Crucially, perplexity is readily available during standard inference and reflects model output distribution changes during fine-tuning regardless of the specific fine-tuning task. This property applies to virtually all decoder-based LLM architectures, making our approach broadly compatible with existing fine-tuning pipelines. We provide a more formal discussion in Section 3.3.

Leveraging the perplexity difference filtering criterion, we propose TuneShift-KD, an automated mechanism that distills specialized knowledge from a source fine-tuned model to a target pre-trained model without full access to the specialized training data set. TuneShift-KD implements the idea by using a few fine-tuning prompts that are representative of the specialized knowledge base to generate similar prompts through an instruction-tuned LLM. We use the phrase "Generate 20 more samples like these 5 [*the 5 sample data*]" to create more training samples in bulk. We discard the bulk-generated responses and keep only the prompts. These prompts are then fed to both the fine-tuned and base models to collect their respective responses. We select prompt-response pairs that demonstrate perplexity differences between the fine-tuned and base model outputs. These selected examples are iteratively added to our training pool and used to generate additional prompts, creating a self-expanding collection of relevant training data. Our solution only requires access to the fine-tuned model, the base model weights, and a few representative examples of the fine-tuning data. For typical PEFT implementations where low-rank adaptors are kept separate from base weights, these requirements are readily satisfied. We also demonstrate that in the absence of the source base model, another general LLM can serve as a substitute.

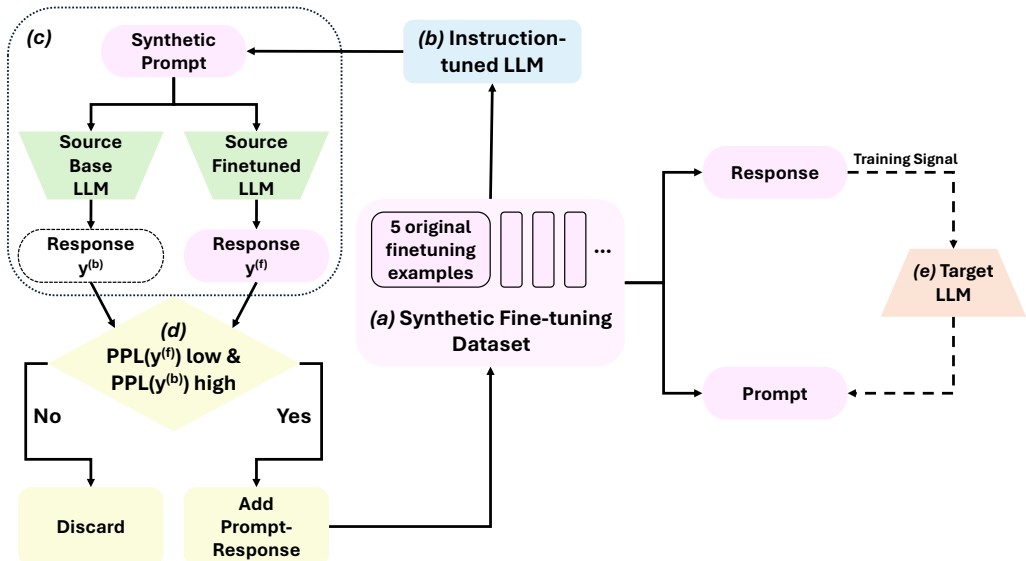

Figure 1: **TuneShift-KD Framework for Specialized Knowledge Transfer.** This figure illustrates our approach to transferring specialized knowledge without requiring full access to original fine-tuning data. **(a)** The Synthetic Fine-tuning Dataset begins with just 5 seed examples from the original fine-tuning data and expands through our iterative process. **(b)** An instruction-tuned LLM generates similar prompts based on the prompt pattern "Generate 20 more samples like these 5 [*the 5 sample data*]". **(c)** Each synthetic prompt is fed to both the Source Base LLM and Source Fine-tuned LLM to obtain paired responses $\mathbf{y}^{(b)}$ and $\mathbf{y}^{(f)}$. **(d)** Our key filtering mechanism selects prompts where the fine-tuned model shows high confidence (low perplexity $\mathrm{PPL}(\mathbf{y}^{(f)})$) while the base model shows low confidence (high perplexity $\mathrm{PPL}(\mathbf{y}^{(b)})$), indicating the prompt targets specialized knowledge. Qualified prompt-response pairs are added to the synthetic dataset, while others are discarded. **(e)** The curated synthetic dataset is used to perform knowledge distillation, transferring specialized capabilities to the Target LLM without requiring original training data.

We empirically validate TuneShift-KD across diverse fine-tuning tasks spanning math reasoning (GSM8K), programming (MBPP), challenging reasoning benchmarks (BBH) Cobbe et al. (2021); Austin et al. (2021); Suzgun et al. (2022). TuneShift-KD consistently improves target model accuracy across different model architectures and outperforms Trans-LoRA despite using only model responses and perplexity difference, without requiring dedicated discriminators trained on the original fine-tuning data. TuneShift-KD remains effective even when the source base model is unavailable, allowing a different generic pre-trained model to serve as a substitute. This architecture flexibility, compatibility with standard fine-tuning pipeline and minimal requirement (only the source fine-tuned model and a generic base model for perplexity comparison) make TuneShift-KD compatible with practical machine learning model deployment scenarios.

In summary, our contributions are:

- We identify perplexity differences between fine-tuned and base model responses as a signal for specialized knowledge, and propose a filtering criterion that requires no external discriminators or architectural constraints.

- We propose TuneShift-KD, an automated method that transfers specialized knowledge from a fine-tuned source model to a target pre-trained model without requiring the original fine-tuning datasets, using iterative synthetic data generation and perplexity-based filtering.

- We demonstrate TuneShift-KD's effectiveness across diverse tasks and model architectures, outperforming Trans-LoRA while maintaining full compatibility with existing fine-tuning pipelines and practical deployment scenarios.

## 2 RELATED WORK

Our work relates to several important research directions in LLMs, including knowledge distillation, synthetic data generation, and data filtering with LLMs. Most closely related is existing research on transferring specialized knowledge from one model to another, which we discuss thoroughly in Section 2.2. Due to space limitations, we defer the discussion of knowledge distillation and synthetic data generation in LLMs to Appendix A.1 and Appendix A.2, respectively. TuneShift-KD uses standard prompt-based synthetic data generation and the negative log-likelihood objective during knowledge distillation.

### 2.1 DATA FILTERING AND PERPLEXITY

Data filtering is a crucial component of the synthetic data generation process. It can be used to improve the quality and relevance of the generated data. There are many different data filtering methods. One category of these methods is classifier-based filtering that trains classifiers to filter out toxic or non-factual samples Kruschwitz & Schmidhuber (2024); Ren et al. (2025). There is a plethora of heuristic-based filtering methods that include human-designed filtering metrics ranging from simple keywords, to template and regex, to higher-level policy-driven checks, such as JSON schema validation, token-length bounds, and fuzzy-deduplication rules Belavadi & Others; Mangalam et al. (2023); Ziegler et al. (2024). Chain-of-Verification utilizes a multi-stage independent response generation for cross-checking Dhuliawala et al. (2023). Crowdsourcing or human-in-the-loop filtering has also been applied Kang et al. (2024). The closest to our work is statistical filtering that leverages perplexity to perform the filtering, which we elaborate on further next.

Perplexity has been used as a straightforward metric to measure LLM confidence and performance Brown et al. (2020). Perplexity has also been used in the data filtering process. For example, *Perplexed by Perplexity* demonstrates that one can train a small LLM to select the data samples at the right perplexity level (dataset dependent), improving the performance of the final LLM training with reduced computational cost. Superfiltering is another technique that uses a small model to select samples that exhibit high perplexity even given the instruction (prompt), and uses these samples to fine-tune a larger model. ScalingFilter Li et al. (2024a) leverages the difference in perplexity between two LLMs of different sizes trained on the same data to identify high-quality text and yields better zero-shot performance when used to filter pretraining data. Iter et al. showed that for in-context demonstrations, selecting examples that reduce the perplexity of the output is an ideal way to improve inference performance with in-context demonstrations Iter et al. (2023). The idea that in-distribution data can have low perplexity has been explored in membership inference attacks. For example, He et al. (2025) and Puerto et al. (2024) assume that in-distribution members have low perplexity, and use this assumption to infer membership. Fu et al. (2024) is another work that leverages the probabilistic variation (similar to perplexity difference in principle) between the member and non-member data to infer membership.

In summary, while leveraging perplexity threshold or perplexity difference has been used in filtering (pre-)training data, improving inference performance and inferring membership, our work is the first to systematically apply the perplexity filter (in the form of perplexity difference filter) to enable specialized knowledge distillation, outperforming existing methods in this task setting while being fully compatible with existing fine-tuning processes. We discuss the theoretical motivation for using the perplexity difference filter to identify the specialized knowledge in Section 3.3. We note that TuneShift-KD is orthogonal to many other techniques that perform data filtering without using perplexity, and those methods could be used for additional verification with human experts Kang et al. (2024) or external information Lupidi et al. (2024) as needed.

### 2.2 TRANSFER OF SPECIALIZED KNOWLEDGE

Transferring the knowledge contained within the fine-tuned component of one model to another has received limited attention. Trans-LoRA is the work with a task setting closest to ours: transferring the specialized knowledge from the source model to a new target model via knowledge distillation Wang et al. (2024). In order to identify fine-tuning relevant knowledge, Trans-LoRA trains dedicated discriminators that can answer the question "Is the above question from *NAME* dataset?". Training such a discriminator requires access to the original fine-tuning data and modifications to all PEFT

processes. The discriminator is an LLM in itself, and separate discriminators need to be trained for different datasets. During the knowledge distillation, the discriminator becomes a data filter, allowing only synthetic data that is classified as *from the dataset* to contribute to the KD process. Such discriminators not only require additional computation to train and storage to keep, but most importantly, are unlikely to be granted by a party that deems the original data not suitable for sharing. In contrast, our solution is compatible with the standard fine-tuning process and also shows improved accuracy compared to Trans-LoRA. In the absence of an open-source implementation of Trans-LoRA, qualitative comparison with specific data samples is difficult. However, we argue that our perplexity difference filtering is a more straightforward method to identify the knowledge that has been acquired by the source model during the fine-tuning process. We compare against Trans-LoRA and demonstrate that our solution leads to higher accuracy for the target model.

LoRA-X is another work that has a similar objective but with a different approach Farhadzadeh et al. (2025). LoRA-X directly transfers the specialized knowledge by copying the LoRA weights from the source model to the target model. Their solution can perform this operation in a data-free manner. However, the alignment of the subspace between the source and target models is a key requirement for the effectiveness of such direct transfer. Empirically, this requires the source and target models to be from the same model family. For example, LoRA-X demonstrated its effectiveness on language tasks by showing little degradation in performance by transferring the LoRA component from a TinyLlama 3T model to TinyLlama 2.5T model, which are highly similar variants from the same model family Zhang et al. (2024). From its main experiments on image diffusion model LoRA transfer, it is observed that the transfer quality severely degrades when the models are not from the same family. In that case, fine-tuning the target model with original fine-tuning data is required to maintain satisfactory performance. A few other works concurrent to us also applied a similar approach to LoRA-X for knowledge transfer, also noting requirements for either architecture family consistency or identical attention mechanisms and activation functions Li et al. (2025); Xia et al. (2025). Compared to these approaches that attempt direct LoRA weight alignment and copying, TuneShift-KD is both more general and complementary. It is more general by operating effectively across different model families without architectural similarity requirements. It is complementary by generating synthetic data that removes LoRA-X's dependency on original fine-tuning datasets when working with dissimilar models.

## 3    METHOD

In this section, we describe our method TuneShift-KD in detail. Figure 1 provides a comprehensive overview of our approach. Our method operates by generating a synthetic fine-tuning dataset that effectively captures the specialized knowledge embedded in the source model. We begin by sampling a minimal seed set of 5 examples from the available fine-tuning data (randomly sampled from the training portion of the dataset). These examples serve as demonstrations for an instruction-tuned LLM, which we prompt to generate 20 similar synthetic examples per seed using the template: "Generate 20 more samples like these 5 [*the 5 sample data*]". This is an iterative process, in which we use the generated synthetic examples to generate more examples.

### 3.1    KNOWLEDGE DISTILLATION

First, we introduce our notations. Starting with the source base model $\mathcal{M}_B$ with parameter $\theta_B$, a source fine-tuned model $\mathcal{M}_F$ with parameter $\theta_F = \theta_B + \phi_F$ has been fine-tuned through a PEFT method on a task-specific dataset $\mathcal{D}$ of prompt–response pairs, where $\phi_F$ is the low-rank adaptor weights. Our goal is to transfer the fine-tuned knowledge of $\mathcal{M}_F$ to a target model $\mathcal{M}_T$. This can be done through knowledge distillation, i.e., fine-tuning parameter $\theta_T$ of the target model $\mathcal{M}_T$ to match the outputs of the source fine-tuned model with parameters $\theta_F$. Specifically, we tune with a sequence-level objective of minimizing the negative log-likelihood of output sequences from the source model:

$$\mathcal{L}(\mathbf{y} \mid \theta_T) = -\sum_{i=1}^{N} \log p_T\big(y_i \mid y_{<i}, x; \theta_T\big) \tag{1}$$

where $\mathbf{y} = (y_1, \ldots, y_N)$ is an output token sequence sampled from the fine-tuned source model's conditional distribution $p_F(\cdot \mid x; \theta_F)$ and $p_T(y_i \mid y_{<i}, x; \theta_T)$ is the fine-tuned target's predicted

probability of token $y_i$ for a given input prompt $x$. In the absence of the original dataset, a critical challenge in this distillation process is obtaining a suitable set of prompts that effectively capture the specialized knowledge of the fine-tuned source model. We discuss this further in the next section.

## 3.2 PERPLEXITY DIFFERENCE FILTERING

Given the same prompt $x$, we get the response $\mathbf{y}^{(f)} \sim p_F(\cdot \mid x; \theta_F)$ from the fine-tuned source model, and the response $\mathbf{y}^{(b)} \sim p_B(\cdot \mid x; \theta_B)$ from the base source model. We then compute the perplexity of both responses under the fine-tuned model:

$$\mathrm{PPL}(\mathbf{y}^{(f)} \mid x; \theta_F) = \exp\big(-\ell(\mathbf{y}^{(f)} \mid x; \theta_F)\big), \tag{2}$$

$$\mathrm{PPL}(\mathbf{y}^{(b)} \mid x; \theta_F) = \exp\big(-\ell(\mathbf{y}^{(b)} \mid x; \theta_F)\big). \tag{3}$$

where

$$\ell(\mathbf{y} \mid x; \theta_F) = \frac{1}{N} \sum_{i=1}^{N} \log p_F(y_i \mid y_{<i}, x; \theta_F) \tag{4}$$

is the per-token average log-likelihood computed using the fine-tuned model parameters $\theta_F$. Calculating perplexity consistently with the fine-tuned model is a deliberate decision, as the fine-tuned model is the most authoritative source on determining the quality of the response in the absence of the original fine-tuning data. Lastly, we select prompt–response pairs where $\mathrm{PPL}(\mathbf{y}^{(f)} \mid x; \theta_F) < \tau \leq \mathrm{PPL}(\mathbf{y}^{(b)} \mid x; \theta_F)$ with $\tau$ a threshold parameter (set to 1.5 by default in our experiments).

## 3.3 PERPLEXITY DIFFERENCE FILTERING AND SPECIALIZED KNOWLEDGE IDENTIFICATION

We now provide a theoretical justification for why perplexity-difference filtering identifies specialized knowledge from the fine-tuned model's perspective. Since the source fine-tuned model $\mathcal{M}_F$ differs from the base model $\mathcal{M}_B$ only through gradient updates on the fine-tuning dataset $\mathcal{D}$, any systematic change in output distributions should stem from the fine-tuning learning process. In the following discussion, we omit the $\theta_F$ and $\theta_B$ references, and replace with $p_F$, $p_B$, $\mathrm{PPL}_F$ for clarity.

Let $p_F(\cdot \mid x)$ and $p_B(\cdot \mid x)$ denote the next-token conditional distributions for a prompt $x$. Given the same $x$, draw two independent responses $\mathbf{y}^{(f)} \sim p_F(\cdot \mid x)$ and $\mathbf{y}^{(b)} \sim p_B(\cdot \mid x)$. The per-token average log-likelihood under the fine-tuned model and its perplexity are given by

$$\ell_F(\mathbf{y} \mid x) = \frac{1}{N} \sum_{i=1}^{N} \log p_F(y_i \mid y_{<i}, x), \qquad \mathrm{PPL}_F(\mathbf{y}) = \exp\big(-\ell_F(\mathbf{y} \mid x)\big). \tag{5}$$

The log-likelihood margin under the fine-tuned scorer is

$$m_F(x) = \ell_F(\mathbf{y}^{(f)} \mid x) - \ell_F(\mathbf{y}^{(b)} \mid x) = -\log \mathrm{PPL}_F(\mathbf{y}^{(f)}) + \log \mathrm{PPL}_F(\mathbf{y}^{(b)}). \tag{6}$$

A positive margin means the fine-tuned model's sample lies in a higher-density region under $p_F$ than the base model's sample.

Taking expectation over the draws $\mathbf{y}^{(f)} \sim p_F$ and $\mathbf{y}^{(b)} \sim p_B$ yields

$$\mathbb{E}\big[m_F(x)\big] = \underbrace{\mathbb{E}_{\mathbf{y} \sim p_F} \ell_F(\mathbf{y} \mid x)}_{-H(p_F)} - \underbrace{\mathbb{E}_{\mathbf{y} \sim p_B} \ell_F(\mathbf{y} \mid x)}_{-H(p_B, p_F)} = H(p_B, p_F) - H(p_F). \tag{7}$$

With $H(p_B, p_F) = H(p_B) + \mathrm{KL}(p_B \| p_F)$, we have

$$\mathbb{E}\big[m_F(x)\big] = (H(p_B) - H(p_F)) + \mathrm{KL}(p_B \| p_F). \tag{8}$$

**Entropy drop** $H(p_B) - H(p_F)$: This captures sharpening. Fine-tuning often concentrates probability mass on fine-tuning–relevant examples, reducing entropy relative to the base model. When this term is positive, fine-tuned samples carry higher per-token log-likelihood under $p_F$, reflecting the updated concentration learned during fine-tuning. If prompts are irrelevant to the fine-tuned behavior, this term will be small.

**Distributional shift** $\mathrm{KL}(p_B \| p_F)$: This measures how much the base distribution disagrees with the fine-tuned one. Larger values indicate that the base tends to produce sequences that $p_F$ scores with lower probability, signaling a mismatch in where the two models place probability mass. When prompts are not relevant to the fine-tuning task, the distributional shift should be small.

Together, these terms show that the expected margin $\mathbb{E}[m_F(x)]$ increases only when prompts $x$ lead to response behaviors actually altered by fine-tuning. For prompts irrelevant to $\mathcal{D}$, the entropy drop vanishes and the KL divergence term is small on average, so the expected margin is near zero. For prompts tied to $\mathcal{D}$, both sharpening and shift contribute, making the expected margin positive.

For the main experiments in this paper, we use a small, fixed threshold $\tau$:

$$\mathrm{PPL}_F(\mathbf{y}^{(f)}) < \tau \le \mathrm{PPL}_F(\mathbf{y}^{(b)}) \implies m_F(x) > 0 \text{ and } m_F(x) > \log\left(\frac{\tau}{\mathrm{PPL}_F(\mathbf{y}^{(f)})}\right). \quad (9)$$

This combines fine-tuned–model preference (the fine-tuned sample beats the base) with an absolute quality gate (the fine-tuned sample lies in a high-density region under $p_F$). A small $\tau$ (e.g., 1.5) keeps examples where the fine-tuned model is confident in its own response while the base model's response is less probable under $p_F$. In Table 8, we also evaluate alternative filters (tighter absolute thresholds and a ratio-based rule) and observe only small differences relative to $\tau = 1.5$, showing that the robustness of our perplexity-difference filter is not tied to the specific thresholding mechanics.

### 3.4 ITERATIVE DATA GENERATION

Our synthetic data generation process begins with a seed set of 5 examples randomly sampled from the available fine-tuning data. Using these examples as demonstrations, we prompt an instruction-tuned LLM (GPT-4o) to generate synthetic examples with the prompt template: "Generate 20 more samples like these 5 [*the 5 sample data*]". We discard the generated responses and keep only the prompts, ensuring we do not leverage GPT-4o's responses in our knowledge transfer. These prompts are then fed to both the source fine-tuned and base models to generate their respective responses. We determine whether to keep each prompt-response pair based on the perplexity threshold defined in Section 3.2. Our choice of instruction-tuned LLM differs from that used in Trans-LoRA, which we discuss further in this section.

The filtered prompt-response pairs serve dual purposes: they constitute our synthetic training dataset for knowledge distillation to the target model, and they expand our synthetic data generation pool for subsequent iterations. In each iteration, we randomly sample 5 examples from the current pool. This iterative process continues until our synthetic dataset matches the size of the original fine-tuning dataset. All sequences in this synthetic dataset are then used to train the target model.

In contrast to Trans-LoRA, which uses the target pre-trained model itself for synthetic prompt generation, we found this approach produced insufficient diversity in the prompts. Despite using the same models and attempting to replicate their reported simple prompting strategy (e.g., "Here are 10 examples"), we observed very limited variation in generated samples across multiple iterations. We demonstrate the limited diversity visually in Figure 2. Our efforts to improve diversity through temperature adjustments and more sophisticated prompting techniques yielded minimal improvements that were insufficient to match their reported accuracy. We note that our use of GPT-4o is limited strictly to prompt generation—we neither use its responses nor leverage it for filtering. The bulk generation approach serves to reduce API costs.

Furthermore, we provide additional experiments in Appendix C using the more capable Qwen2.5 Qwen (2024) families as the source and target models. In those experiments, the Qwen2.5 models themselves serve as the instruction-tuned LLM, generating the synthetic prompts, showing that TuneShift-KD does not rely on external instruction-tuned LLMs.

## 4 EVALUATION

### 4.1 EXPERIMENT SETUP

We evaluate our method on a set of popular benchmarks, including BigBench-Hard (BBH) Suzgun et al. (2022), Mostly Basic Python Problems (MBPP) Austin et al. (2021), Grade School Math 8K (GSM8K) Cobbe et al. (2021). BBH is a suite of 23 challenging tasks from the broader BIG-Bench

Table 1: GSM8K benchmark results

| Source | Target | Trans-LoRA | | | Ours | | |
|---|---|---|---|---|---|---|---|
| | | Source LoRA | Target (no LoRA) | Target LoRA | Source LoRA | Target (no LoRA) | Target LoRA |
| Llama2-7B | Llama2-13B | 19.64% | 28.86% | 30.70% (↑ 1.84%) | 19.48% | 27.89% | 30.12% (↑ 2.23%) |
| Gemma-2B | Gemma-7B | 14.94% | 40.64% | 44.58% (↑ **3.94**%) | 15.20% | 40.0% | 44.80% (↑ **4.80**%) |

Table 2: MBPP benchmark results (standard evaluation)

| Source | Target | Trans-LoRA | | | Ours | | |
|---|---|---|---|---|---|---|---|
| | | Source LoRA | Target (no LoRA) | Target LoRA | Source LoRA | Target (no LoRA) | Target LoRA |
| Llama2-7B | Llama2-13B | 27.2% | 37.1% | 39.7% (↑ 2.6%) | 28.2% | 36.9% | 40.2% (↑ **3.3**%) |
| Gemma-2B | Gemma-7B | 41.1% | 37.9% | 50.0% (↑ 12.1%) | 40.9% | 37.8% | 51.3% (↑ **13.5**%) |

Table 3: BBH benchmark results

| Source | Target | Trans-LoRA | | | Ours | | |
|---|---|---|---|---|---|---|---|
| | | Source LoRA | Target (no LoRA) | Target LoRA | Source LoRA | Target (no LoRA) | Target LoRA |
| Llama2-7B | Llama2-13B | 43.32% | 37.85% | 43.41% (↑ 5.56%) | 42.02% | 38.04% | 44.92% (↑ **6.88**%) |
| Gemma-2B | Gemma-7B | 31.84% | 37.75% | 43.61% (↑ 5.86%) | 31.01% | 38.15% | 45.09% (↑ **6.94**%) |

evaluation, selected for their difficulty and where prior language models failed to outperform average human raters. MBPP comprises 974 crowd-sourced Python programming problems designed to test code synthesis capabilities on basic programming constructs and standard library usage. GSM8K consists of 8,500 linguistically diverse grade school math word problems that require multi-step arithmetic reasoning to solve. In order for a fair comparison with Trans-LoRA, we similarly evaluate BBH, GSM8K using the Language Model Evaluation Harness framework Gao et al. (2024), and evaluate MBPP using Evalplus Liu et al. (2023). Consistent with Trans-LoRA, we demonstrate our method on the Llama-2 Touvron et al. (2023) and Gemma Gemma Team (2024) families of LLM models. Specifically, we use Llama-2 in the 7B, 13B model sizes and Gemma in the 2B, 7B sizes.

## 4.2 RESULTS

We compare our target model's performance before and after our knowledge distillation process with Trans-LoRA across different datasets in Tables 1, 2, and 3. We note that we use the same models and datasets as Trans-LoRA. However, due to inherent noise in the token sampling process and lack of open-source implementations for Trans-LoRA, we are not able to perfectly reproduce the same accuracy as Trans-LoRA's results that should be independent of the synthetic data generation process, e.g. the "Source LoRA" and "Target (no LoRA)" columns. However, our results are within a small margin of Trans-LoRA's for those.

We note that the key metric to focus on the target model's accuracy improvement as a result of the LoRA knowledge distillation process (column "Target LoRA ↑"). Across all three datasets (GSM8K, MBPP and BBH), our solution shows greater accuracy improvement compared to Trans-LoRA in the target model after the knowledge distillation. We note that such accuracy improvement is achieved while our solution is directly applicable to the standard fine-tuning pipeline, without the need to have access to the original fine-tuning dataset to train auxiliary modules (such as the discriminator).

Table 4: Generic pre-trained model as base substitutes

| Dataset | Source Base | Source LoRA | Target | Target (no LoRA) | Target LoRA | Our Acc Increase |
|---------|-------------|-------------|--------|------------------|-------------|------------------|
| GSM8K | Gemma-2B | Llama2-7B | Gemma-7B | 40.0% | 43.82% | ↑3.82% |
| MBPP | Llama2-7B | Gemma-2B | Gemma-7B | 37.8% | 51.2% | ↑13.4% |

Example-based qualitative comparisons to reveal why our method has improved accuracy are difficult as Trans-LoRA does not have an open-source implementation. However, we hypothesize the following reasons for our higher accuracy improvement:

**TuneShift-KD's data filtering process is beyond simple discrimination** Trans-LoRA employs a binary discriminator that classifies examples as either from the fine-tuning data or not. While our perplexity difference filter serves a similar purpose, it operates with greater nuance: it selects examples where the fine-tuned model responds with high confidence (low perplexity) while the base model struggles (high perplexity), directly targeting specialized knowledge that *the fine-tuned model has acquired*. Our approach automatically eliminates two categories of less useful examples: those where both models demonstrate high confidence (indicating knowledge already presented in pre-training) and those where both models show uncertainty (suggesting examples too difficult for the fine-tuned model to learn). This strategic filtering focuses specifically on knowledge that the fine-tuned model has successfully internalized, removing the fine-tuned model's responses that would contribute minimally to a target model's learning process.

**Trans-LoRA's discriminator faces inherent objective conflicts** Trans-LoRA adopts a GAN-inspired approach but with a critical deviation: rather than utilizing the generator, it preserves the discriminator after training. This discriminator is explicitly trained to distinguish between authentic fine-tuning data ("real") and synthetically generated examples ("fake"). Yet during inference, the system must use this same discriminator to evaluate synthetic examples—the very category it was optimized to identify as inauthentic. This training-inference objective contradiction likely constrains the synthetic data generator to converge on an artificially narrow subset of examples that can circumvent the discriminator's filters. Given that GANs are inherently susceptible to mode collapse Arjovsky et al. (2017), and considering our observation that the target model already demonstrates limited prompt generation diversity, Trans-LoRA's methodology appears inclined to produce a restricted range of synthetic training examples.

### 4.3 RESULTS WHEN SOURCE BASE MODEL IS UNAVAILABLE

We demonstrate additional results on the target model performance when the actual base model of the fine-tuned model is not available in Table 4. We show that in this situation, another generic pre-trained model that wasn't fine-tuned on the specialized knowledge can be used to serve as a substitute for the actual base model. On the GSM8K dataset, using the Llama2-7B as the substitute (in the absence of access to the Gemma-2B base), resulted in an accuracy increase of 3.82% of the target model—a slight drop from the 4.80% increase if the Gemma-2B base was available (as shown in Table 1). On MBPP, the accuracy drop was even smaller, at 0.1% (compared to Table 2). These results demonstrate that, while having access to the base model helps, it is not a strict requirement to ensure effectiveness of our data generation and filtering process.

## 5 CONCLUSION

We presented TuneShift-KD, a method for transferring specialized knowledge between language models without requiring access to original fine-tuning data. By leveraging perplexity differences between fine-tuned and base models, our approach identifies and filters synthetic examples that effectively capture specialized knowledge. Experiments across GSM8K, MBPP, and BBH tasks demonstrate improvements over existing methods, with our solution's primary advantage being its broad applicability and compatibility with standard fine-tuning pipelines. TuneShift-KD addresses practical scenarios where original training data is unavailable due to privacy concerns or commercial restrictions, enabling efficient knowledge transfer between different model architectures.

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

We present additional related work, results and ablations to the main experiments in the appendix. Section A reviews related work on knowledge distillation in LLMs and synthetic data generation with LLMs. Section B provides additional results for different datasets using the same experimental setting as in the main paper. Section C presents the results of TuneShift-KD when applied to the Qwen2.5 architecture. Section D explores different perplexity filter thresholds and filtering mechanisms. Section E visualizes prompt diversity comparisons between our approach and related works. Section F shows example prompts and responses generated by the fine-tuned and base models. We discuss ethics and privacy concerns in Section G, experimental settings and reproducibility in Section H, and LLM usage in Section I.

## A  ADDITIONAL RELATED WORK

### A.1  KNOWLEDGE DISTILLATION FOR LLMS

Knowledge distillation (KD) Hinton et al. (2015) has been an important technique for distilling knowledge from a large source model to a smaller target model. For classification and regression tasks, KD techniques include logit-based distillation, feature-based distillation Romero et al. (2014), and response-based distillation Park et al. (2019). These KD techniques are widely used in transferring the knowledge from a large source model to a smaller target model for improved inference efficiency, but can generally work regardless of the model sizes Furlanello et al. (2018); Zhang et al. (2017). Encoder-only language models that focus on contextual embedding (classification) tasks use similar KD techniques, examples include DistilBERT, TinyBERT, and MiniLM Sanh et al. (2019); Jiao et al. (2019); Wang et al. (2020). However, decoder-only generative language models require different treatment of KD techniques.

Knowledge distillation for decoder-only autoregressive language models usually operates with a sequence-level objective Kim & Rush (2016); Zhong et al. (2024). The target model's objective typically involves either minimizing the negative log-likelihood of source model-sampled output sequences or minimizing the KL divergence between its token logits and the source model's Wen et al. (2023); Kim et al. (2021). Researchers have also explored generalized divergence objectives, such as f-divergence Wen et al. (2023). Important applications include compressing larger source models to smaller target models for inference efficiency Timiryasov & Tastet (2023), and transferring capabilities from stronger proprietary models to open source models Chen et al. (2024). Some works have also focused on skill-specific distillations, like chain-of-thought distillation for enhanced reasoning Li et al. (2023) or alignment distillation for human value preference alignment Gu et al. (2025). Self-distillation is another application that supports continual learning and regularization by iteratively refining the same model Wang et al. (2021).

While our work and existing KD works both seek to transfer knowledge from one model to another, we focus on transferring the specialized knowledge in the absence of the full fine-tuning data, for which standard KD techniques are not readily applicable.

### A.2  SYNTHETIC DATA GENERATION WITH LLMS

Large Language Models (LLM) readily serve as effective tools for synthetic data generation due to their ability to generate coherent, contextually relevant text. A simple approach involves utilizing few-shot in-context learning, after which LLMs can generate task-aligned examples that broadly reflect the structure and style of the target data Brown et al. (2020). Prompt-based generation techniques leverage simple or complex prompts to guide LLMs in producing diverse, task-specific synthetic examples with minimal examples Liu et al. (2021a). This approach has been shown to produce high-quality synthetic question–answer pairs that improve downstream model performance on tasks such as domain-specific QA Schmidt et al. (2024). Synthetic data generation in general benefits a wide range of language-related tasks. For example, in neural machine translation, back-translation generates synthetic parallel data that, when combined with authentic parallel data, leads to consistent improvements in translation quality on standard WMT benchmarks Sennrich et al. (2015). For programming tasks, LLMs generate synthetic code snippets and unit tests that facilitate large-scale code modeling and analysis Schäfer et al. (2023). In privacy-sensitive domains such as healthcare, synthetic data with differential privacy protections maintains utility while safeguarding individual record confidentiality Pang et al. (2024).

Our method follows the established simple prompt-and-few-shot framework, using LLMs to generate question variants based on a small set of examples from the fine-tuning dataset. This approach could potentially benefit scenarios where original data is unavailable due to privacy constraints, though in the current work we demonstrate its efficacy using various standard fine-tuning datasets.

Table 5: MMLU benchmark results

| | | Trans-LoRA | | | | Ours | | | |
|---|---|---|---|---|---|---|---|---|---|
| Source | Target | Source LoRA | Target (no LoRA) | Target LoRA | Acc Increase | Source LoRA | Target (no LoRA) | Target LoRA | Acc Increase |
| Llama2-7B | Llama2-13B | 45.89% | 53.72% | 55.09% | 1.37% | 47.19% | 53.13% | 53.62% | 0.49% |
| Gemma-2B | Gemma-7B | 42.34% | 60.45% | 61.23% | 0.78% | 41.39% | 61.62% | 62.38% | 0.76% |

Table 6: MBPP benchmark results (strict evaluation)

| | | Trans-LoRA | | | | Ours | | | |
|---|---|---|---|---|---|---|---|---|---|
| Source | Target | Source LoRA | Target (no LoRA) | Target LoRA | Acc Increase | Source LoRA | Target (no LoRA) | Target LoRA | Acc Increase |
| Llama2-7B | Llama2-13B | 25.0% | 31.7% | 34.4% | 2.7% | 25.7% | 31.6% | 33.9% | 2.3% |
| Gemma-2B | Gemma-7B | 33.9% | 32.1% | 40.6% | 8.5% | 32.4% | 40.7% | 46.8% | 6.1% |

## B  RESULTS FOR ADDITIONAL DATASETS

In this section, we present additional results for different datasets using the same experiment setting as in the main paper.

**MMLU**   In Table 5, we present additional results for the MMLU dataset. In general, we found that the accuracy increase from fine-tuning is very small, consistent with observations from Trans-LoRA. For example, during our testing, the Llama2-13B model's accuracy after fine-tuning increases by only 0.49% (from 53.13% to 53.62%). For the Gemma-7B model, the increase is similarly small at 0.76%. We believe this occurs because the MMLU dataset is highly fact-based, and the training split provides little transferable knowledge to the evaluation split. Therefore, whether the target model successfully distills knowledge from the source fine-tuned model has minimal impact on the target model's accuracy increase. The target model likely relies on knowledge obtained during pre-training rather than from distillation.

**Strict MBPP+**   In Table 6, we present additional results for the strict version of the MBPP dataset, referred to as "MBPP+" Austin et al. (2021). To the best of our knowledge, this corresponds to the same strict MBPP+ dataset used by Trans-LoRA. However, we observe substantial discrepancies between our accuracy and Trans-LoRA's reported results. Most notably, Trans-LoRA reports 32.1% accuracy for their "Target (no LoRA)" baseline, while we achieve 40.7% accuracy for our "Target (no LoRA)" baseline using the Gemma-7B model. This 8.6 percentage point difference represents a significant gap in the evaluation of the same pre-trained model without any fine-tuning.

We believe these discrepancies likely stem from differences in evaluation settings. Unfortunately, due to the lack of an open-source implementation of Trans-LoRA, we could not fully resolve these differences. Nevertheless, we note two important observations. First, when comparing accuracy improvements rather than absolute values, our gains are comparable to those reported by Trans-LoRA. Combined with our superior performance on other datasets where results are more reproducible, we believe our solution achieves higher accuracy overall, and the specific discrepancy in this strict MBPP+ setting is due to implementation-related factors. Second, in Appendix E, we provide t-SNE visualizations of MBPP prompts generated using Llama2 (following Trans-LoRA's reported setting) versus GPT-4o (our setting). These visualizations demonstrate that our prompts exhibit significantly greater diversity. The limited diversity in Trans-LoRA's prompts likely restricts knowledge extraction from the fine-tuned model, making it unlikely that Trans-LoRA's approach can achieve comparable target accuracy improvements.

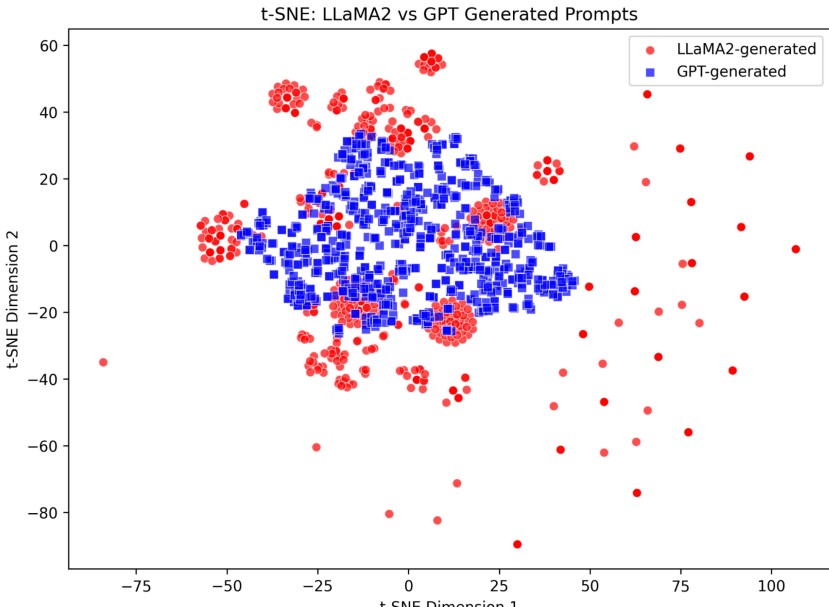

Figure 2: t-SNE visualization of MBPP prompt embeddings generated by Llama2-13B versus GPT-4o. Embeddings computed using the MPNet encoder model demonstrate the superior diversity of GPT-4o generated prompts.

## C RESULTS FOR ADDITIONAL ARCHITECTURES FOR TUNESHIFT-KD

Table 7: TuneShift-KD results using Qwen models as source and target

| Dataset | Source | Target | Source LoRA | Target (no LoRA) | Target LoRA | Acc Increase |
|---------|--------|--------|-------------|------------------|-------------|--------------|
| GSM8K | Qwen2.5-7B Instruct | Qwen2.5-14B Instruct | 69.4% | 43.9% | 79.6% | 35.7% |
| MBPP | Qwen2.5-7B Instruct | Qwen2.5-14B Instruct | 52.8% | 42.0% | 60.1% | 18.1% |

We provide additional evaluations with the Qwen2.5 architectures in Table 7. The Qwen target model achieved significant accuracy improvements (35.7% on GSM8K and 18.1% on MBPP) through knowledge transfer using TuneShift-KD. Unlike the Llama and Gemma experiments where we used GPT-4o for prompt generation, we used the source Qwen2.5 model itself as the instruction-tuned LLM for synthetic prompt generation. This demonstrates that TuneShift-KD can operate without external prompt generation models when the source model has sufficient prompt generation capabilities. The limitations of prompt generation diversity in Llama and Gemma models were discussed in Section 3.4 and visualized further in Figure 2.

## D RESULTS FOR DIFFERENT PERPLEXITY FILTER SETTINGS

Our perplexity filtering mechanism demonstrates robustness across different threshold configurations. While our main experiments use a symmetric threshold of 1.5 (retaining examples where fine-tuned model perplexity $< 1.5$ and base model perplexity $> 1.5$), we evaluate several alternative settings in Table 8.

Setting (a) uses a lower symmetric threshold of 1.3, resulting in only a 0.1% accuracy decrease compared to our default configuration. Setting (b) employs asymmetric thresholds (fine-tuned $< 1.2$, base $> 1.6$), creating a wider perplexity gap requirement. This avoids retaining borderline cases where models show minimal differences (e.g., fine-tuned perplexity of 1.49 vs. base perplexity of 1.51), but yields virtually identical performance (–0.02% change). Setting (c) uses a ratio-based approach, retaining examples where the base model perplexity is at least 1.5 times the fine-tuned model perplexity. This dynamic threshold slightly improves accuracy by 0.13%.

Table 8: GSM8K performance under different perplexity filtering configurations. **(a)** Symmetric threshold filtering with both models using threshold 1.3. **(b)** Asymmetric threshold filtering with fine-tuned model perplexity $< 1.2$ and base model perplexity $> 1.6$. **(c)** Ratio-based filtering where examples are retained if $1.5 \times$ fine-tuned perplexity $\leq$ base perplexity.

| Filter Setting | Source Model | Target Model | Target (no LoRA) | Target LoRA Acc. | Acc. Change (vs. default) |
|---|---|---|---|---|---|
| **(a)** Threshold = 1.3 | Llama2-7B | Llama2-13B | 27.89% | 30.02% | –0.10% |
| **(b)** Fine-tuned $< 1.2$, Base $> 1.6$ | Llama2-7B | Llama2-13B | 27.89% | 30.10% | –0.02% |
| **(c)** Ratio-based: $1.5\times$ fine-tuned $\leq$ base | Llama2-7B | Llama2-13B | 27.89% | 30.25% | +0.13% |

The consistent performance across these configurations confirms our approach's robustness to specific threshold choices—a valuable property when extensive hyperparameter tuning is impractical due to the absence of validation data from the original fine-tuning process.

Table 9: Comparison of filtering strategies: no filtering vs. alternative filtering criteria

| Dataset | Source | Target | Target (no LoRA) | Low-High Filter (TuneShift-KD) | No Filter Acc | Low-Low Filter Acc | High-High Filter Acc |
|---|---|---|---|---|---|---|---|
| GSM8K | Gemma-2B | Gemma-7B | 40.0% | **44.8**% | 36.7% | 40.2% | 35.4% |
| GSM8K | Llama2-7B | Llama2-13B | 27.9% | **30.1**% | 27.8% | 28.4% | 26.9% |
| MBPP | Gemma-2B | Gemma-7B | 37.8% | **51.2**% | 35.8% | 39.5% | 34.4% |
| MBPP | Llama2-7B | Llama2-13B | 36.9% | **40.2**% | 29.4% | 37.4% | 30.2% |

In Table 9, we compare our perplexity difference filtering (threshold = 1.5) with alternative filtering strategies. **No Filter** applies synthetic data generation and knowledge distillation with a pipeline identical to TuneShift-KD but without any prompt filtering. **Low-Low Filter** selects prompts where both the source base and fine-tuned models exhibit low perplexity (threshold = 1.5). **High-High Filter** selects prompts where both models exhibit high perplexity. Across all datasets and model architectures, TuneShift-KD's low-high perplexity difference filtering achieves the highest target accuracy. The Low-Low filter shows only modest improvements over the not fine-tuned baseline (Target no LoRA), while No Filter degrades target model performance. High-High filtering further reduces accuracy compared to the no filter case. These results demonstrate that TuneShift-KD's filtering criterion successfully identifies beneficial training examples while avoiding harmful ones that would otherwise degrade model performance.

Together, these ablation studies demonstrate that while the specifics of the perplexity difference filtering mechanism has little impact, the application of it is important as the target model accuracy can even degrade when naively applying all synthetic data without filter or selecting high perplexity (confusing) examples.

# E    VISUALIZATION OF PROMPT DIVERSITY

We demonstrate the difference in prompt diversity between our method and Trans-LoRA's approach. We use the MPNet Song et al. (2020) model to generate embeddings for examples produced by Llama2-13B (following Trans-LoRA's reported setting) and GPT-4o (our setting). MPNet was also used in Trans-LoRA's paper for embedding analysis. We then visualize the t-SNE plots of the embeddings generated from both model outputs.

As shown in Figure 2, GPT-4o generates significantly more diverse prompts than Llama2-13B. The Llama2-13B prompts cluster into a few small, tightly-grouped regions, with highly similar prompts within each cluster. In contrast, GPT-4o prompts exhibit much broader distribution across the embedding space. This limited diversity in Llama2-13B's outputs likely impedes the target model's ability to learn from a varied set of examples that would optimally extract knowledge from the fine-tuned source model.

In Figures 3 and 4, we show the perplexity distributions of prompts generated by the fine-tuned (teacher) and base Llama2-7B models. We observe that the teacher model's perplexity is much smaller compared to the base model for both datasets. For the MBPP examples, we found that although many of the teacher's generated examples have perplexity above 1.5, they should not be retained as they do

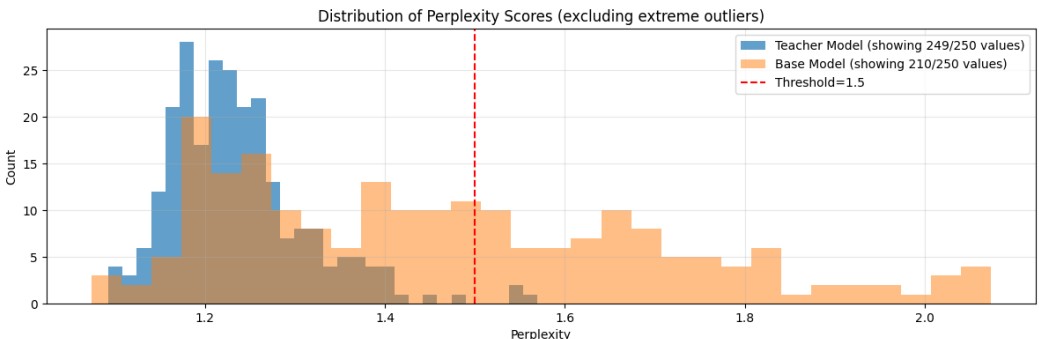

Figure 3: Perplexity distribution of GSM8K examples generated by fine-tuned (Teacher) and base Llama2-7B models. Perplexities beyond the rightmost bin are omitted for clarity.

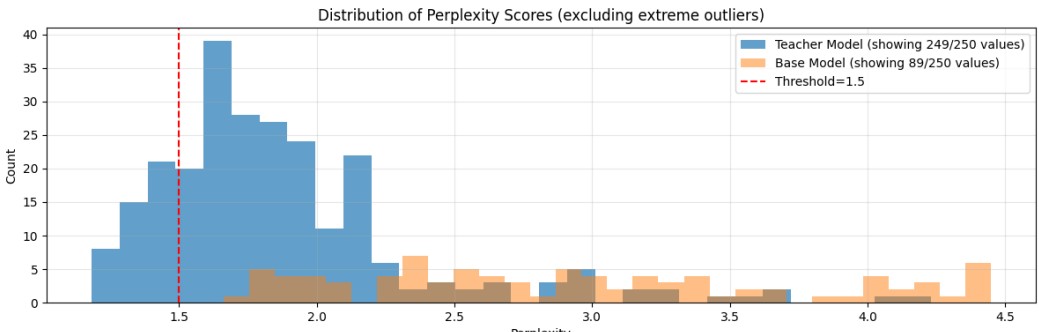

Figure 4: Perplexity distribution of MBPP examples generated by fine-tuned (Teacher) and base Llama2-7B models. Perplexities beyond the rightmost bin are omitted for clarity.

not provide correct task demonstrations. In other words, we found the 1.5 threshold to be a reasonable setting that excludes examples where the teacher model lacks confidence across multiple datasets.

## F  GENERATED EXAMPLES

In this section, we show example prompts and responses generated by the fine-tuned and base models. In our experiments, we observe that many of our generated samples are correctly retained through our filtering process—that is, prompts that lead to technically correct fine-tuned responses and incorrect base responses are kept. Here, we focus on demonstrating a wide range of examples, including potential edge cases, to illustrate both examples that are technically correct and could be retained, as well as those that are technically incorrect and could be filtered out. We note that the final target model accuracy depends on many factors, including data quality, correctness, and diversity, as well as other considerations such as the target model's training dynamics. In general, correctness alone is insufficient for determining data inclusion or exclusion. We comment on the responses' correctness for demonstration purposes only.

## G  ETHICS AND PRIVACY

TuneShift-KD operates within the practical constraint where fine-tuned model weights are accessible but original training data is not. This is a common scenario in real-world ML deployments where service providers receive models for deployment without accessing proprietary datasets, often due to privacy reasons. For example, cloud providers hosting LoRA adapters motivated Trans-LoRA's data-free approach Wang et al. (2024). Secure cloud architectures like AWS Nitro treat model weights and training data as separate assets with different access controls Amazon Web Services (2024). Additionally, the growing "Data-Free Knowledge Distillation" literature specifically targets scenarios where datasets are withheld for privacy or IP reasons Liu et al. (2024; 2021b).

```
{
  "prompt": "Question: Noah bought 5 bundles of balloons, and each
  ↪  bundle had 9 balloons. 7 balloons popped. How many balloons does
  ↪  he still have?\nAnswer:",
  "fine-tuned response": "Noah has 5 x 9 = <<5*9=45>>45 balloons.\n7
  ↪  balloons popped, so he has 45 - 7 = <<45-7=38>>38 balloons
  ↪  left.",
  "base response": "Noah has 8 balloons left."
}
```

Figure 5: GSM8K example generated by fine-tuned and base Llama2-7B, with 1.19 fine-tuned perplexity and 4.22 base perplexity. The fine-tuned response is technically correct, and the base response is incorrect. The example was **kept** through the perplexity filter process.

```
{
  "prompt": "Question: Layla baked 5 trays of muffins. Each tray had 8
  ↪  chocolate muffins and 4 blueberry muffins. She gave away 10
  ↪  muffins. How many does she have left?\nAnswer:",
  "fine-tuned response": "Layla baked 5 x 8 = <<5*8=40>>40
  ↪  muffins.\nShe gave away 10 + 40 = <<10+40=50>>50 muffins.\nSo
  ↪  she has 40 - 50 = <<40-50=10>>10 muffins left.",
  "base response": "Layla has 40 muffins left."
}
```

Figure 6: GSM8K example generated by fine-tuned and base Llama2-7B, with 1.16 fine-tuned perplexity and 3.05 base perplexity. Both fine-tuned and base responses are technically incorrect. The example was **kept** through the perplexity filter process.

```
{
  "prompt": "Question: Carter baked 48 muffins. He put them into boxes
  ↪  with 6 muffins each. Then he ate 2 muffins. How many full boxes
  ↪  remain?\nAnswer:",
  "fine-tuned response": "Carter baked 48 / 6 = <<48/6=8>>8 boxes of
  ↪  muffins.\nHe ate 2 / 6 = <<2/6=1>>1 muffin per box.\nSo, 8 - 1 =
  ↪  <<8-1=7>>7 full boxes remain.",
  "base response": "After Carter baked 48 muffins and put them into
  ↪  boxes with 6 muffins each, he had 8 boxes. If he ate 2 muffins, he
  ↪  has 7 full boxes remaining."
}
```

Figure 7: GSM8K example generated by fine-tuned and base Llama2-7B, with 1.32 fine-tuned perplexity and 1.93 base perplexity. Both responses are technically correct, though they employ different solution methods, resulting in different perplexities. The example was **kept** through the perplexity filter process.

**Privacy properties of TuneShift-KD:** TuneShift-KD is not itself a privacy-preserving method. Our approach accesses information through standard model queries and therefore cannot extract knowledge beyond what the source model already exposes. If a deployed model does not generate sensitive information through normal querying, TuneShift-KD introduces no additional privacy risks. For models trained with differential privacy, querying does not weaken existing guarantees or consume additional privacy budget Yu et al. (2021); Du et al. (2024). However, if the source model leaks sensitive information through standard prompting, TuneShift-KD may incorporate such information into synthetic training data. Thus, source model providers should ensure their models do not contain or produce sensitive information through prompting. Ensuring this is an orthogonal problem to TuneShift-KD usage.

```
{
"prompt": "Write a function to generate a list of n Fibonacci
↪   numbers.",
"fine-tuned response": "def fibonacci(n):\n    if n == 0:\n
↪   return 0\n    if n == 1:\n        return 1\n    return
↪   fibonacci(n−1) + fibonacci(n−2)",
"base response": "def generate_fibonacci(n):\n    a, b = 0, 1\n
↪   result = []\n    for _ in range(n):\n        result.append(a)\n
↪   a, b = b, a + b\n    return result"
}
```

Figure 8: MBPP example generated by fine-tuned and base Gemma-2B, with 1.28 fine-tuned perplexity and 1.45 base perplexity. The fine-tuned response is correct, and the base response is incorrect. The example was **kept** through the perplexity filter process.

```
{
"prompt": "Write a function to interleave two strings of the same
↪   length.",
"fine-tuned response": "def interleave(s1, s2):\n    return ''.join(i
↪   for i in (s1 + s2))",
"base response": "def interleave_strings(a, b):\n    return ''.join(x
↪   + y for x, y in zip(a, b))"
}
```

Figure 9: MBPP example generated by fine-tuned and base Gemma-2B, with 1.67 fine-tuned perplexity and 1.68 base perplexity. Both responses are technically correct and very similar, resulting in almost identical perplexities. The example was **removed** through the perplexity filter process.

```
{
"prompt": "Write a function to reverse a list without using built-in
↪   reverse().",
"fine-tuned response": "def reverseList(myList):\nreturn
↪   myList[::-1]",
"base response": "def reverse_list(lst):\n    return lst[::-1]"
}
```

Figure 10: MBPP example generated by fine-tuned and base Gemma-2B, with 2.01 fine-tuned perplexity and 1.66 base perplexity. The fine-tuned response has an incorrect indentation, resulting in a rare case where the teacher's solution is incorrect but the base solution is correct. The example was **removed** through the perplexity filter process.

In summary, TuneShift-KD operates within existing privacy boundaries established by source models without introducing novel privacy risks beyond standard model querying. We recommend implementing proper safeguards at the source model level; however, TuneShift-KD users should also be aware of potential privacy risks if the source model was not properly safeguarded.

## H  REPRODUCIBILITY

### H.1  EXPERIMENTAL SETTINGS

We trained six models (Llama2-7B Chat, Llama2-13B Chat, Gemma-2B, Gemma-7B, Qwen2.5-7B Instruct, Qwen2.5-14B Instruct) on two AMD Instinct MI210 GPUs using low-rank adaptation (LoRA). LoRA adapters with rank $r = 8$, scaling factor $\alpha = 16$, and dropout rate $p = 0.05$ were inserted into the query and value projection layers of each multi-head attention block. All models were trained for 20 epochs using the AdamW optimizer with a peak learning rate of $2 \times 10^{-5}$ and

linear decay. Fine-tuning was performed in 16-bit floating-point precision (FP16). These settings apply to both source model fine-tuning and target model knowledge distillation. We include the source code in the supplementary material and will open-source it upon publication.

## H.2 COMPUTATIONAL COST AND SCALABILITY

Our method's computational overhead remains practical compared to standard fine-tuning. For GSM8K with the Llama family, fine-tuning takes approximately 30 minutes, while data generation and perplexity filtering require 2 hours and 1 hour, respectively, on two AMD Instinct MI210 GPUs. This pattern holds across datasets and models, with TuneShift-KD requiring roughly $6\times$ the time of direct fine-tuning. Based on typical GPU rental costs of approximately \$2.00 per hour for equivalent hardware DeepSeek-AI (2024), the total cost for GSM8K knowledge transfer is approximately \$14.00.

Following Trans-LoRA's experimental setting, we use 250 final synthetic examples for target model training, with approximately 80% filtered out during perplexity screening (this ratio varies by model and dataset but remains relatively stable). We generate prompts in batches of 20. Given an average prompt length of 50 words in GSM8K, this requires fewer than 100,000 tokens, costing under \$1 in GPT-4o API calls Microsoft Azure (2025). Our generation and filtering pipeline is inherently parallelizable, allowing speedup with additional GPUs for larger datasets. Given our near data-free setting, this computational cost represents a reasonable trade-off for enabling knowledge transfer without original data access.

# I LLM USAGE

LLMs were an inherent part of TuneShift-KD research. Different models were used for training, fine-tuning, synthetic data generation, and filtering. The detailed usage has been discussed thoroughly throughout the paper.

The authors also used LLMs for polishing the writing of this paper.

