# OpenReview forum: "TuneShift-KD: Knowledge Distillation and Transfer for Fine-tuned Models"
_ICLR.cc/2026/Conference — Submitted to ICLR 2026_

### Official Review · Reviewer_QRr3 · 2025-10-26

**Soundness:** 3
**Presentation:** 3
**Contribution:** 2
**Rating:** 4
**Confidence:** 5

**Summary:**

The paper introduces TuneShift-KD, a method for transferring specialized knowledge from a fine-tuned model to a different pre-trained model without access to the original fine-tuning dataset. It addresses scenarios where domain-specific data cannot be shared due to privacy or commercial restrictions.
This approach uses a small set of seed examples to generate synthetic prompts, then applies a filtering criterion based on perplexity differences between the fine-tuned model and its base model.
The difference in confidence between the fine-tuned model (high) and its base model (low) is used to select prompts indicative of specialized knowledge.
These filtered examples form a synthetic dataset used for knowledge distillation to the target model.
This method is automated and architecture-agnostic, requiring no discriminators.
Experiments on GSM8K, MBPP, and BBH benchmarks show that TuneShift-KD improves target model accuracy compared to prior approaches like Trans-LoRA, even when the original base model is unavailable.

**Strengths:**

## Originality
This paper introduces a new problem formulation: transferring specialized capabilities from one model variant to another without access to the original fine-tuning data.
The originality lies in using perplexity differences between fine-tuned and base models as a signal to identify domain-specific knowledge,
combined with iterative synthetic query generation.
While knowledge distillation and synthetic data generation are established ideas, their combination with perplexity-based filtering for data-free transfer across model families is novel.

## Quality
The proposed methodology is clearly described and includes theoretical reasoning for why perplexity differences indicate specialized knowledge, supported by experiments on multiple benchmarks.
However, the theoretical foundation is limited to information divergence arguments, and the approach remains heavily dependent on the quality of the fine-tuned model and seed examples.
Experimental results show consistent improvements over prior work, but comparisons are constrained by the lack of open-source baselines.

## Clarity
This paper is well-structured, with clear explanations of the problem, method, and evaluation.
Figures and tables illustrate the pipeline and results effectively.
The iterative process and filtering criterion are explained in detail, though some sections assume familiarity with PEFT and LoRA, which may challenge readers without prior knowledge.

## Significance
This work addresses a practical and increasingly common scenario in real-world deployments where fine-tuning data cannot be shared.
Its compatibility with standard model adaptation workflows and ability to operate across different model architectures make it relevant for industry applications.
As it removes a key limitation of prior approaches that required original datasets or structural, its theoretical novelty is limited, but the method offers strong practical utility in real-world deployment scenarios.

**Weaknesses:**

## Dependence on Fine-Tuned Model Quality
This approach assumes that the fine-tuned model accurately represents domain-specific knowledge.
If the fine-tuned model contains errors or biases, these will transfer to the target model.
The paper acknowledges this but does not propose mechanisms to mitigate it.
A possible improvement is integrating external validation or correctness checks during filtering, such as leveraging trusted knowledge sources or consistency verification methods.

## Limited Theoretical Foundation
The justification for using perplexity differences relies on entropy and KL divergence arguments, which are presented briefly and lack formal proofs or deeper analysis.
Strengthening the theoretical basis, for example by connecting the filtering criterion to generalization bounds or information-theoretic measures, would improve rigor and credibility.

## Model and Data Dependency
Performance depends heavily on the fine-tuned model, seed examples, and the target model’s capacity.
The current paper does not explore strategies to reduce this dependency. Introducing adaptive sampling that considers target model weaknesses or curriculum-based distillation could make the method more robust and efficient.

## Synthetic Prompt Generation Risks
The method uses few-shot prompting to generate synthetic prompts, but correctness and diversity are not guaranteed. While diversity is discussed, there is no systematic evaluation of prompt quality. Adding quantitative measures of diversity and correctness, or incorporating filtering based on semantic similarity and factuality, would strengthen reliability.

## Evaluation Scope
Experiments focus on three benchmarks and two model families, with additional tests on Qwen.
However, the currrent paper does not include ablations on initial example set size, generation strategy, or alternative filtering metrics beyond perplexity.
Broader evaluation, including low-resource domains or multilingual settings, would clarify generalizability.

## No Comparison with Target-Adaptive Filtering (e.g., weakness-aware data selection)
This paper does not consider approaches that select examples based on the target model’s weaknesses, which could improve efficiency. Including such a baseline or discussing its feasibility would provide a more complete picture of the design space.

**Questions:**

Q1. How does the proposed method ensure that the synthetic prompts generated from a small seed set are representative of the original fine-tuning domain? Could you provide quantitative evidence of prompt diversity and domain relevance beyond t-SNE visualization?

Q2. The filtering criterion relies on perplexity differences computed using fine-tuned models. How sensitive is the method to the choice of threshold τ, and have you considered adaptive or ratio-based thresholds tied to distributional properties rather than fixed values?

Q3. Since fine-tuned models may contain errors or biases, what mechanisms could be integrated to prevent propagating incorrect knowledge during distillation? Have you explored external validation or correctness checks in the filtering process?

Q4. Why was the decision made to use the base model for filtering rather than the target model? Would incorporating the target model’s weaknesses into the selection process improve efficiency and accuracy? Have you considered or tested target-adaptive filtering strategies?

Q5. The theoretical justification for perplexity difference is based on entropy and KL divergence arguments. Could you provide more formal analysis or empirical evidence that this criterion consistently correlates with domain-specific knowledge across diverse tasks?

Q6. How does the method handle cases where the fine-tuned model’s specialized knowledge overlaps significantly with general knowledge? Does the filtering process risk discarding useful examples or retaining trivial ones?

Q7. Synthetic prompt generation uses external instruction-tuned LLMs or the source model itself. How do you ensure factual correctness and avoid introducing noise? Have you evaluated the impact of prompt quality on final distillation performance?

Q8. Experiments focus on GSM8K, MBPP, and BBH. Could you clarify why these benchmarks were chosen and whether the method generalizes to other domains such as multilingual tasks or low-resource settings?

Q9. The paper claims compatibility with standard fine-tuning pipelines. Could you elaborate on practical deployment considerations, such as computational cost for large-scale models and scenarios where external LLMs for prompt generation are unavailable?

Q10. How does TuneShift-KD compare to approaches that directly transfer LoRA weights when models share similar architectures? Would combining weight transfer with your synthetic data approach yield better results?

---

> ### Author Response · Authors · 2025-12-02
>
> We thank the reviewer for recognizing the originality of our perplexity-based filtering approach, the clarity of our presentation, and the practical significance of our method for real-world deployment scenarios. We address the weaknesses and questions below.
>
> **W1: [External Verification]** We acknowledge that errors or biases in the fine-tuned model will transfer to the target model. However, the problem of LLMs producing confident yet incorrect responses remains a significant unsolved challenge in LLM research [1]. Various approaches have emerged, including: (i) probing internal representations to estimate correctness [2]; (ii) sampling multiple reasoning paths and selecting the most consistent answer [3]; (iii) self-evaluation through question-writing and verification [4]; (iv) training models to abstain when uncertain [5]; and (v) verifying answers against external evidence via retrieval-augmented generation [6,7].
> While these techniques can mitigate confident-but-wrong outputs, they are largely orthogonal to TuneShift-KD and sometimes incompatible with our setting. Some require modifying the source model or accessing external data, assumptions we deliberately avoid. Others, though compatible, incur significant computational overhead. In practice, these complementary methods could be layered after our inexpensive perplexity filtering. For example, Chain-of-Verification (CoVe) [4] could verify surviving prompt-response pairs, though at additional computational cost. Combining suitable verification methods with TuneShift-KD could yield more accurate solutions, and this accuracy-computation tradeoff warrants further study.
>
> [1] Yadav et al. "Know Your Limits: A Survey of Abstention in Large Language Models" NAACL 2024
>
> [2] Xiong et al. "Robust Confidence Estimation in LLMs through Internal States" ACL 2024
>
> [3] Manakul et al. "SelfCheckGPT: Zero-Resource Black-Box Hallucination Detection for Generative Large Language Models" EMNLP 2023
>
> [4] Dhuliawala et al. "Chain-of-Verification Reduces Hallucination in Large Language Models" arXiv 2023
>
> [5] Amayuelas et al. "Large Language Models Must Be Taught to Know What They Don't Know" ACL 2024
>
> [6] Wei et al. "Retrieval Augmented Fact Verification by Synthesizing Contrastive Examples" ACL 2024
>
> [7] Chen et al. "Faithfulness-Aware Uncertainty Quantification for Fact-Checking the Output of Retrieval Augmented Generation" arXiv 2025
>
>
>
> **W2: [Theoretical Foundation]** We discuss the theoretical foundation in Section 3.3, clearly deriving and demonstrating that entropy drop and distributional shift are the two components our mechanism identifies. These are precisely the information-theoretic measures the reviewer requested.
>
> **W3: [Model and Data Dependency]** Our solution is designed to transfer knowledge from the source fine-tuned model to the target model. Correcting errors in LLM outputs is itself ongoing research and remains an unsolved problem beyond the scope of our current work. Our focus is on transferring the source fine-tuned model's specialized knowledge to the target model. Correctness guarantees, to the extent possible, should be ensured in the source fine-tuned model, which is beyond the scope of our problem setting.
>
> **W4: [Correctness and Diversity]** As discussed in W1 and W3, correctness could be improved using other tools, but achieving perfect correctness remains an important open problem in LLM research and is beyond our scope. Regarding diversity, we have included extensive discussion throughout the paper and supplementary material.
>
> **W5: [Evaluation Scope]** We examine different perplexity difference mechanisms in Table 8. From the theoretical discussion in Section 3.3, we clearly demonstrate how this is directly related to model distribution differences. Our perplexity difference mechanism is grounded in first principles. While alternative mechanisms may exist, they may target different aspects of the model beyond the scope of this research. Our evaluation uses relatively few resources due to the limited initial data available. Since our mechanism only requires perplexity scores, it should extend to multilingual settings where such information is also available.

---

> > ### Author Response · Authors · 2025-12-02
> >
> > **Q1: [Fine-tuning domain]** As discussed in Section 3.3, the perplexity difference must necessarily stem from model behavior changes induced by the fine-tuning process. Our perplexity difference criterion is precisely designed to identify such changes. We provide quantitative evidence in Table 8 that data selected with our method substantially improves target model performance on the fine-tuning evaluation domain.
> >
> > **Q2: [Filtering criterion]** The choices have been ablated in Table 8. We tested ratio-based thresholds. Adaptive thresholds could be interesting but would be very data-dependent. We found our method is robust to different specific choices of threshold.
> >
> > **Q3: [Error Mechanism]** As discussed in W1, W3, and W4, our method aims to transfer knowledge from the source fine-tuned model, which may contain proprietary information that cannot be evaluated or verified without access to the original data. Such access may not be available in our setting. However, if verification data is available, external verification tools can be used in combination with our method. Moreover, the better solution is to ensure the source fine-tuned model produces correct outputs, which makes this a general problem of ensuring LLMs produce only correct answers, an ongoing area of research in itself.
> >
> > **Q4: [Base model for filtering]** As discussed in Section 3.3, our goal is to identify changes induced by the fine-tuning process applied to the base model. This information is only available by analyzing the base model using perplexity difference filtering, not the target model.
> >
> > **Q5: [Perplexity Difference Justification]** The analysis and derivation are grounded in KL divergence from first principles. We demonstrate empirically the differences in model responses and how our method identifies them through qualitative examples in Figures 5 through 10.
> >
> > **Q6: [Knowledge Overlaps]** Even in scenarios with overlap, our method continues to identify differences between the models, which is our design target. This may mean more generated examples are filtered out, but we still identify the remaining model differences.
> >
> > **Q7: [Factual Correctness]** See W1, W3, W4, Q3.
> >
> > **Q8: [Experiment choice]** The datasets were chosen such that our experimental setting is consistent with Trans-LoRA for direct comparison. Our method is generally very low-resource, requiring only a few original examples. As discussed in W5, for multilingual tasks, perplexity information is directly available, and our method should readily work with such settings.
> >
> > **Q9: [Deployment Consideration]** We discuss computational costs in detail in Appendix H.2. When external LLMs are unavailable, as long as the source fine-tuned model can generate diverse prompts, we can use it directly, as demonstrated in the Qwen-based experiments.
> >
> > **Q10: [Combination with other methods]** Direct LoRA weight transfer methods work in very limited settings where models are from the same family. In these limited scenarios, combining their method (first copying weights) with ours (then performing knowledge distillation with samples selected by our method) could further improve performance. However, since weight transfer within the same model family is highly limited and specific, our method remains more robust and applicable in general cases.

---

### Official Review · Reviewer_UVzK · 2025-10-28

**Soundness:** 2
**Presentation:** 2
**Contribution:** 2
**Rating:** 2
**Confidence:** 3

**Summary:**

This paper presents a method called TuneShift-KD, which aims to transfer knowledge from a fine-tuned model to a new target model when the original fine-tuning data are unavailable. The authors compare the perplexity differences between the fine-tuned model and its base model on generated outputs to identify samples that purportedly contain “specialized knowledge,” which are then used for subsequent distillation. The overall reasoning of the paper relies on a heuristic assumption that perplexity differences reflect knowledge differences; however, this assumption lacks rigorous theoretical grounding and comprehensive empirical validation. The core procedure depends on several empirically chosen hyperparameters and the stable behavior of generative models, which limits robustness. Furthermore, the experimental design shows constraints in dataset selection and baseline comparisons, making it difficult to demonstrate the generality and reliability of the proposed approach. Overall, while the work attempts to address a practical problem, the current experiments and analysis do not sufficiently substantiate its effectiveness or novelty.

**Strengths:**

1. The authors address a genuine and relevant challenge in model transfer and knowledge distillation — how to extract specialized knowledge from an existing fine-tuned model when the original fine-tuning data are unavailable. This problem has practical significance in real-world scenarios, particularly where data access is limited or restricted by compliance constraints.
2. TuneShift-KD avoids the need for additional discriminators or manual labeling by relying on perplexity-based filtering and iterative data generation to construct distillation samples. The overall framework is relatively simple and can be integrated into existing fine-tuning pipelines with minimal modification.

**Weaknesses:**

1. Lack of theoretical grounding: The core assumption—that perplexity differences can effectively represent knowledge differences between models—has no solid theoretical justification. The authors provide only heuristic reasoning without statistical significance analysis or ablation comparing alternative indicators such as KL divergence or output diversity.

2. Overly heuristic and weakly interpretable method: The key filtering mechanism of TuneShift-KD depends on an empirically chosen threshold (e.g., τ = 1.5), yet the paper does not analyze sensitivity to this parameter or explain why it generalizes across tasks and model scales. This raises concerns about reproducibility and robustness.

3. Limited experimental scope and shallow validation: The evaluation is restricted to a few benchmarks (GSM8K, MBPP, BBH) that mainly test small-scale reasoning or code generation. There is no validation on more complex or safety-critical tasks (e.g., multi-turn dialogue, factual knowledge distillation, or alignment), undermining the claim of general knowledge transfer.

4. Unfair or insufficient baselines: Comparisons with existing approaches such as Trans-LoRA and LoRA-X lack strict control over model size, training steps, and computational budgets, which weakens the credibility of the reported performance improvements.

5. Missing ablation and failure analyses: The paper does not disentangle the contribution of each component in TuneShift-KD or analyze failure cases, leaving unclear whether improvements arise from the core idea or from secondary factors such as data regeneration or repeated distillation.

6. Marginal and statistically unstable improvements: On some datasets (e.g., MMLU), the gains are negligible or inconsistent, yet the paper does not report confidence intervals or variance across runs, making the results statistically unconvincing.

7. Limited novelty: The approach essentially combines known ideas from knowledge distillation and synthetic data generation, with the main modification being a new heuristic for sample selection. It lacks substantial theoretical or algorithmic innovation.

**Questions:**

1. Please elaborate on the theoretical foundation of the assumption that “perplexity difference reflects knowledge difference.” Is there a more rigorous explanation from an information-theoretic or distributional perspective? If so, could the authors provide mathematical reasoning or additional supporting experiments in the appendix?
2. The key threshold in TuneShift-KD (e.g., τ = 1.5) appears to be chosen empirically. How is this value determined? Does the performance vary significantly under different thresholds? A systematic sensitivity analysis would help demonstrate the robustness of the method.
3. The comparisons with methods such as Trans-LoRA and LoRA-X lack details about computational budget, training epochs, and learning rates. Could the authors clarify these settings to ensure that the reported results are reproducible under equivalent conditions?
4. Please specify under what conditions TuneShift-KD performs best, and in which scenarios (e.g., fact-heavy tasks or open-domain QA) it fails. Providing representative failure cases and analyses would help readers understand the boundaries of the method.
5. Beyond perplexity difference, have the authors explored alternative indicators such as KL divergence, cross-entropy margin, or token-level consistency? If so, please include comparative results to justify the choice of perplexity as the main criterion.
6. How does TuneShift-KD fundamentally differ from existing knowledge distillation or synthetic data generation approaches? The authors are encouraged to more clearly articulate the novel contribution and situate the method within the broader research landscape.

---

> ### Author Response · Authors · 2025-12-02
>
> We thank the reviewer for recognizing the practicality and generality of our method. We address the weaknesses and questions below.
>
> **W1: [Lack of theoretical grounding]** Our perplexity filtering method demonstrates strong empirical performance, especially when compared to the no-filter baseline (see Table 9). We provide detailed theoretical justification in Section 3.3, explaining which differences between source models our method identifies. The theoretical discussion centers on KL divergence and demonstrates how our perplexity difference criterion captures KL divergence differences between base and fine-tuned model distributions. We also extensively discuss our model choice to ensure output diversity, which is an important component of the overall pipeline. In summary, while we rely on KL divergence and output diversity in our pipeline, they serve as the theoretical foundation rather than direct indicators for the filtering process.
>
> **W2: [Filtering Mechanism]** We demonstrate empirically that our method is robust to a wide range of perplexity difference choices in Table 8, including different values of τ and slightly altered mechanisms. This shows our method generalizes across different tasks and models. In Figures 3 and 4, we demonstrate the separation between teacher and base model perplexities using histogram bins, clearly showing the distributional differences between models. Our results can be reproduced using the supplementary code included with the submission.
>
> **W3: [Evaluation Scope]** Our evaluation spans different domains, including factual knowledge, coding, and mathematical reasoning. While post-training can employ various techniques including alignment and behavior tuning, which could be interesting components to transfer, our evaluated tasks already cover a wide range of specialized capabilities and knowledge types.
>
> **W4: [Baselines]** We selected baselines that address the same problem setting as ours. We used similar training datasets, epochs, and numbers of synthetic examples as Trans-LoRA whenever possible. Our experimental settings are detailed in Appendix H.1, and computational resources are discussed in Appendix H.2. Trans-LoRA should be more computationally expensive than our method due to discriminator training. Our costs for synthetic data generation and distillation are comparable, and computing perplexity from generated data has trivial computational cost. As discussed extensively in the related work section, LoRA-X works only when source and target models are from identical model families, a restriction our method does not have. Although LoRA-X may have faster runtime in that limited setting, our solution is substantially more general.
>
> **W5: [Ablations and Failure Analysis]** Our ablations are presented in Appendices C through F and Figures 3-10, examining different aspects of our design choices and their impact on performance. We demonstrate a failure case in Figure 10 where the source fine-tuned model provides inadequate output. We also decouple the data generation process from the filtering mechanism in Table 9, where applying no filter results in much lower target model performance, demonstrating the effectiveness of our filtering mechanism.
>
> **W6: [Marginal Improvements]** We demonstrate consistent gains in target model performance across all datasets. The magnitude of performance improvements depends on task difficulty, base model robustness, and many factors outside the scope of our method.
>
> We provide statistical significance analysis below. We conducted additional runs with different random seeds on the GSM8K and MBPP datasets. Results show minimal variance across runs, demonstrating that our method consistently and reliably improves target model accuracy.
>
> | Dataset | Source LoRA | Target      | Target LoRA (Original) | Target LoRA (New 1) | Target LoRA (New 2) | Average with Error Bar |
> |---------|-------------|-------------|------------------------|---------------------|---------------------|------------------------|
> | GSM8K   | Llama2-7B   | Llama2-13B  | 30.1%                  | 30.0%               | 29.9%               | 30.1% ±0.058%          |
> | GSM8K   | Gemma-2B    | Gemma-7B    | 44.8%                  | 44.6%               | 45.1%               | 44.8% ±0.15%           |
> | MBPP    | Llama2-7B   | Llama2-13B  | 40.2%                  | 40.3%               | 40.0%               | 40.2% ±0.088%          |
> | MBPP    | Gemma-2B    | Gemma-7B    | 51.2%                  | 50.9%               | 51.2%               | 51.1% ±0.10%          |

---

> > ### Author Response · Authors · 2025-12-02
> >
> > **W7: [Novelty]** Our work is the first to demonstrate that perplexity difference filtering can be used to identify and extract specialized information from fine-tuned models. While our method uses synthetic data generation and knowledge distillation, our goal is not to develop new data generation or distillation algorithms. Regarding theoretical and algorithmic innovation, we clearly demonstrate in Section 3.3 why perplexity difference filtering is a robust and general method for detecting model distribution shifts from a theoretical perspective.
> >
> >
> > **Q1: [Theoretical Foundation]** Our theoretical foundation is clearly presented in Section 3.3, with detailed derivations and explanations of what differences our filtering method identifies.
> >
> > **Q2: [Thresholds]** We have tested different thresholds and mechanisms  in our ablation study (Table 8).
> >
> > **Q3: [Baselines]** See W4.
> >
> > **Q4: [Ablations and Failure Analysis]** See W5.
> >
> > **Q5: [Alternative mechanisms]** Our mechanism targets the indicators suggested by the reviewer. As extensively discussed in Section 3.3, our filtering mechanism precisely targets KL divergence between models. Cross-entropy margins can be realized by choosing perplexity difference thresholds with gaps, as demonstrated in Table 8(b). Token-level consistency is also directly related to perplexity difference. This demonstrates the flexibility and robustness of our approach in identifying key information from the models.
> >
> > **Q6: [Difference from KD and Data Generation]** As discussed in W7, our work focuses on the data sampling mechanism and how perplexity difference can be an effective tool for identifying model differences, supported by theoretical grounding. It is not intended to be a new method for knowledge distillation or synthetic data generation.

---

### Official Review · Reviewer_aF6d · 2025-10-29

**Soundness:** 2
**Presentation:** 2
**Contribution:** 2
**Rating:** 2
**Confidence:** 3

**Summary:**

The paper proposes TuneShift-KD, a novel approach that automatically distills specialized knowledge from a fine-tuned model to a target model. The method only uses a few examples that represent this specialized information. The key idea is that the specialized knowledge can be identified through perplexity differences between the base and fine-tuned models. The method uses this idea to create a synthetic dataset of training samples and performs SFT on these. The results demonstrate that TuneShift-KD captures a larger share of the fine-tuned knowledge than prior methods, while being easier to deploy in practice.

**Strengths:**

- The perplexity difference criterion is intuitive. Prompts where the fine-tuned models are confident but base models struggle can capture specialized knowledge.
- Unlike Trans-LoRA, TuneShift-KD requires no discriminator, which makes the standard fine-tuning process simpler and more practical.
- The method shows accuracy gains over Trans-LoRA in GSM8K, MBPP, and BBH.
- The model is highly automatic, can transfer information across different architectures, and works without the exact base model.

**Weaknesses:**

- As the authors acknowledge, perplexity/likelihood-based selection of training samples is a well-known technique, known already 15 years ago (see e.g., [1]). Even in an LLM-based distillation context, log-likelihood/entropy-based methods have been used recently (see e.g., [2, 3]). This work is clearly part of the same family of data selection methods, limiting the novelty.
- My interpretation of the results is that the main performance driver compared to Trans-LoRA is the diversity of the prompts. It is unclear if the difference is due to the discriminator vs perplexity-based selection of synthetic data or simply the generator model for prompts.
- The paper frames itself as trying to solve the distillation of domain-specific, specialized knowledge. However, the successful benchmarks are BBH, math, and coding, which can be seen as a big mismatch with the stated motivation. Because these benchmarks are so generic, it is easy to see how Gpt-4o (or Qwen as in Appendix C) can generate relevant prompts. However, if the domain is truly out-of-distribution (such as internal company data), it is unclear if a generic target model can generate relevant prompts even in a few-shot setting to achieve a sufficient factual coverage. The poor performance in MMLU (Table 5) indicates that this issue might be serious. This makes a method like LoRA-X much more attractive, despite its significant drawbacks.
- The assumption that a subset of the data, but not all of it, can be kept to act as seed samples is arguably quite restrictive. How realistic is it, really, especially assuming that you nevertheless retain access to the model weights?
- No error bars, variances, or multiple seeds. Are the results statistically significant?
- Lack of information about the data for fine-tuning the source model.
- The perplexity difference between the fine-tuned model and the base model could also be partially due to the style of the fine-tuning data (or some other reason causing a generic distribution shift), which could harm the perplexity threshold-based selection in a more realistic setting.

[1] Moore, R. C., & Lewis, W. (2010). Intelligent selection of language model training data. In Proceedings of the ACL 2010 conference short papers (pp. 220-224).

[2] Li, J., Nag, S., Liu, H., Tang, X., Sarwar, S., Cui, L., ... & Tang, J. (2024). Learning with less: Knowledge distillation from large language models via unlabeled data. arXiv preprint arXiv:2411.08028.

[3] Liu, J., Zhang, C., Guo, J., Zhang, Y., Que, H., Deng, K., ... & Zheng, B. (2024). Ddk: Distilling domain knowledge for efficient large language models. Advances in Neural Information Processing Systems, 37, 98297-98319.

**Questions:**

- If you argue that your perplexity-based prompt selection method is meaningfully different from Li et al.'s teacher confidence/student uncertainty for sample selection (see [2] above), this sample selection method should be a baseline.
- Would it be possible to run experiments with, for instance, LLama2 for prompt generation, to truly isolate whether the perplexity-based selector is meaningfully different from the discriminator in terms of performance (or if it's simply a question of efficiency & ease of implementation)?
- Can you run experiments on synthetic, truly out-of-distribution data and compare your method to Trans-LoRA, LoRA-X, and standard SFT on the original data (as an upper bound) under that setting?
- Trans-LoRA has released at least some code as part of the paper supplementary material (https://openreview.net/forum?id=c3Pakdyi3t). Did you consider this when claiming there's no open-source implementation of Trans-LoRA or is there some issue with this code?
- You write that GANs suffer from mode collapse, which affects Trans-LoRA. That seems unlikely, as the generator in Trans-LoRA is the target model M_t (with only instruction tuning). On the other hand, the objective mismatch makes more sense as an explanation, but could you provide empirical evidence in support of that being the issue (vs the generator model)?

---

> ### Author Response · Authors · 2025-12-02
>
> We thank the reviewer for recognizing the intuitive and automatic nature of our method, as well as the performance and practical benefits over Trans-LoRA. We address the weaknesses and questions below.
>
> **W1: [Perplexity-based selection]** We recognize that perplexity is an important and widely-used metric for understanding LLM behavior. However, leveraging perplexity differences to identify specialized knowledge between base and fine-tuned models is novel to the best of our knowledge.
>
> **W2: [Performance driver]** Unfortunately, we cannot reproduce Trans-LoRA's behavior (we discuss this further below). However, generating high-quality, diverse prompts without access to the original training data is precisely the challenge in data-absent settings. Our perplexity difference filtering ensures prompt quality and must be combined with generation diversity to produce meaningful prompts and responses. Without proper perplexity-based filtering (sampling everything indiscriminately), prompts may still be diverse since they are selected with fewer constraints, but distillation can become harmful. As demonstrated in Table 9 (Appendix D), unfiltered synthetic data degrades target model performance.
>
> **W3: [Domain knowledge]** While the target model may lack domain knowledge, our pipeline leverages the source fine-tuned model for prompt generation when it has sufficient generation diversity (e.g., Qwen models). Since the source fine-tuned model was trained on domain-specific data, it has encountered these types of prompts during fine-tuning and can generate domain-relevant examples.
>
> **W4: [Seed sample availability]** The scenario where a few seed examples are available while the full training dataset is not is realistic and common in practice. In industry deployments, service providers often receive fine-tuned model weights for deployment without accessing the complete proprietary training datasets. However, obtaining a handful of representative examples (5 in our case) is far less restrictive than accessing thousands of training samples. These seed examples can come from public documentation, API usage examples, or minimal data sharing agreements that allow limited samples for validation purposes while protecting the bulk of proprietary data.
> This separation between model weights and full training data is well-documented. Cloud providers hosting LoRA adapters motivated Trans-LoRA's data-free approach [1]. Hardware vendors like NVIDIA TensorRT [2] and Intel Neural Compute Stick [3] compile customer models without dataset access. AWS Nitro [4] architectures treat model weights and training data as separate assets with different access controls. The "Data-Free Knowledge Distillation" literature [5, 6] specifically addresses scenarios where datasets are withheld for privacy or IP reasons. TuneShift-KD's minimal seed requirement (5 examples) makes it practical even when comprehensive data sharing is prohibited, addressing a realistic constraint in real-world ML deployments.
> [1] Wang et al. "Trans-LoRA: Towards Data-Free Transferable Parameter Efficient Finetuning" Neurips 2024
> [2] NVIDIA TensorRT Documentation: "Overview" https://docs.nvidia.com/deeplearning/tensorrt/
> [3] Intel Corporation "Intel Neural Compute Stick 2" Press Release 2018
> [4] AWS "A Secure Approach to Generative AI with AWS" Blog Post 2024
> [5] Liu et al. "Small Scale Data-Free Knowledge Distillation" CVPR 2024
> [6] Zhang et al. "Data Free Knowledge Distillation: A Survey" Nature Scientific Reports 2024
>
>
> **W5: [Statistical Significance]** We provide statistical significance analysis below. We conducted additional runs with different random seeds on the GSM8K and MBPP datasets. Results show minimal variance across runs, demonstrating that our method consistently and reliably improves target model accuracy.
>
> | Dataset | Source LoRA | Target      | Target LoRA (Original) | Target LoRA (New 1) | Target LoRA (New 2) | Average with Error Bar |
> |---------|-------------|-------------|------------------------|---------------------|---------------------|------------------------|
> | GSM8K   | Llama2-7B   | Llama2-13B  | 30.1%                  | 30.0%               | 29.9%               | 30.1% ±0.058%          |
> | GSM8K   | Gemma-2B    | Gemma-7B    | 44.8%                  | 44.6%               | 45.1%               | 44.8% ±0.15%           |
> | MBPP    | Llama2-7B   | Llama2-13B  | 40.2%                  | 40.3%               | 40.0%               | 40.2% ±0.088%          |
> | MBPP    | Gemma-2B    | Gemma-7B    | 51.2%                  | 50.9%               | 51.2%               | 51.1% ±0.10%          |

---

> > ### Author Response · Authors · 2025-12-02
> >
> > **W6: [Fine-tuning source model]** The hyperparameters for training the source model are detailed in Appendix H.1. We apply standard LoRA training procedures to ensure realistic experimental settings.
> >
> > **W7: [Perplexity difference source]** We acknowledge that perplexity differences may be induced by the style of the fine-tuning data. However, if such differences are introduced by the source fine-tuned model, knowledge distillation should faithfully transfer those changes to the target model. This is why we chose perplexity difference as our criterion: it targets distributional shifts generally, regardless of the specific characteristics of the dataset being used.
> >
> > **Q1: [Comparison with Li et al.]** There are two substantial differences that make our settings incomparable. First, Li et al. do not identify differences between a source fine-tuned model and its base model, but rather work with a generic teacher model. Second, they assume access to unlabeled text data from the target domain, assuming the textual data itself is available. In contrast, our pipeline is designed for scenarios where the training data is unavailable. Direct comparison would be unfair since Li et al.'s method has access to substantial domain data (albeit unlabeled), while our method operates with only a few seed examples.
> >
> > **Q2: [Llama2 for prompt generation]** We did experiment with this initially. However, performance degrades significantly due to the very limited diversity in Llama2's prompt generation. Although our perplexity difference filter can select high-quality prompts in this setting, the lack of diversity prevents us from meaningfully extracting sufficient information from the source model. This limitation should equally apply to Trans-LoRA: regardless of filtering quality, prompts must be reasonably diverse initially. As discussed extensively in the paper, we do not believe Trans-LoRA's reported performance is realistic due to concerns about both diversity and quality. We further discuss the lack of open-source code for Trans-LoRA below.
> >
> > **Q3: [Out-of-distribution data]** Standard model pre-training procedures do not rely on the evaluation datasets we use (GSM8K, MBPP, BBH), as model developers also evaluate performance on these benchmarks. This is precisely why significant distributional shifts exist between the source base and fine-tuned models. We recognize that experiments with truly private, out-of-distribution data would be valuable. However, we are not able to obtain such data easily due to practical resource and access constraints.
> >
> > **Q4: [Trans-LoRA code]** Trans-LoRA's code from OpenReview contains only entry points: train.py, which requires access to synthetic data for knowledge distillation, and synthesis.py, which requires access to the discriminator to produce synthetic data. However, the discriminator training code and trained weights cannot be found on GitHub or Hugging Face. Trans-LoRA's GitHub repository (https://github.com/raywang4/TransLoRA) remains marked "To be released after approval," and no weights are available on Hugging Face (https://huggingface.co/papers/2405.17258). Effectively, Trans-LoRA's OpenReview code contains only standard LLM data generation and knowledge distillation procedures, which are well-known pipelines. Their key contributions, including modeling, training, and using the discriminator, are completely missing, and we were unable to reproduce their results.
> >
> > **Q5: [Mode collapse]** Trans-LoRA's generator undergoes standard GAN training, which is known to be susceptible to mode collapse [7]. Our empirical observation is that Trans-LoRA's choice of target model, even before fine-tuning, generates prompts with very limited diversity. This alone, regardless of the data selection procedure, makes it difficult to meaningfully obtain a wide range of information from the source fine-tuned model. Further fine-tuning reduces prompt diversity even further. Our discussion of the objective mismatch represents another significant obstacle, even if Trans-LoRA managed to generate reasonably diverse prompts. However, given Trans-LoRA's current state of limited diversity, the discriminator has very few samples to select from initially. It is therefore very difficult to draw conclusions, whether empirically or theoretically, regarding sample quality shifts due to objective mismatch with only a very limited set of samples.
> >
> > [7] Salimans et al., “Improved Techniques for Training GANs,” NeurIPS 2016

---

### Official Review · Reviewer_PJqH · 2025-10-31

**Soundness:** 3
**Presentation:** 3
**Contribution:** 2
**Rating:** 4
**Confidence:** 3

**Summary:**

The paper proposes TuneShift-KD, a method for transferring specialized knowledge from a fine-tuned source model (e.g., LoRA-adapted) to a new target model without access to the original fine-tuning data.
The key insight is to identify “specialized knowledge” regions using perplexity difference between the fine-tuned and base models: prompts where the fine-tuned model is confident (low PPL) but the base model is uncertain (high PPL) are assumed to encode domain-specific expertise.

Using a few seed examples, an instruction-tuned LLM (e.g., GPT-4o) generates synthetic prompts. These are filtered by the perplexity-difference criterion to form a synthetic dataset used for knowledge distillation (via NLL loss) into the target model.
Experiments on GSM8K, MBPP, and BBH benchmarks show consistent gains over Trans-LoRA, a previous data-free transfer method, while being simpler and applicable across model families (LLaMA2, Gemma, Qwen).

**Strengths:**

The paper addresses a practical and emerging challenge, motivated by real deployment constraints (privacy, cloud-hosted models, hardware vendors): transferring LoRA-fine-tuned expertise when the fine-tuning data are unavailable.

The perplexity-difference criterion is intuitive, theoretically grounded (entropy and KL analysis), and easy to implement.

Broad applicability and strong empirical results

**Weaknesses:**

While elegant, the main novelty—using PPL difference as a filter—is incremental compared to prior perplexity-based data filtering

Results are limited to small-/mid-scale models (≤13 B). It is unclear whether the method scales or remains stable for larger modern architectures (e.g., 70 B).

Reported improvements (1–7 pp) are modest and could lie within the noise range of evaluation harnesses, yet statistical significance is not reported.

I think the tasks (GSM8K, MBPP, BBH) are general-domain, not truly “specialized knowledge.” Stronger validation would involve real domain-specific fine-tuning (e.g., medical, legal, or code-domain LoRAs) where data unavailability is realistic.

**Questions:**

How exactly is “specialized knowledge” defined or operationalized? Does it refer to new factual content, new reasoning skills, or simply distributional shift between base and fine-tuned models?

The paper claims improved prompt diversity over Trans-LoRA. Can this be quantified (e.g., via lexical or semantic diversity metrics)?

The paper uses NLL loss on fine-tuned model outputs. Have the authors compared it with logit-based or KL-based distillation (e.g., temperature-scaled soft labels)?

---

> ### Author Response · Authors · 2025-12-02
>
> We thank the reviewer for recognizing the practicality and soundness of the method. We address the weaknesses and questions below.
>
> **W1: [PPL difference as a filter]** The introduction of PPL difference filtering is indeed one of our main contributions, alongside the general practical pipeline. While perplexity has been used in prior work to understand model behavior and output distributions, no existing work has systematically demonstrated how to leverage this metric to identify differences between models for specialized knowledge transfer. Our contribution lies in showing that a widely-accepted metric can be repurposed with theoretical grounding (Section 3.3) to enable effective cross-model knowledge distillation without discriminators or architectural constraints. This combination of simplicity, theoretical justification, and practical effectiveness makes our approach both novel and valuable to the community.
>
> **W2: [Model scale]** Due to resource constraints, we are unable to perform additional experiments on larger-scale models. However, our method demonstrates strong performance across architectures of varying sizes (2B to 14B parameters), and the core criterion in our framework, perplexity difference, is model-size agnostic. Since perplexity computation scales naturally with model size and requires no architectural modifications, our pipeline should operate on larger models without design changes.
>
> **W3: [Statistical Significance]** We provide statistical significance analysis below. We conducted additional runs with different random seeds on the GSM8K and MBPP datasets. Results show minimal variance across runs, demonstrating that our method consistently and reliably improves target model accuracy.
>
> | Dataset | Source LoRA | Target      | Target LoRA (Original) | Target LoRA (New 1) | Target LoRA (New 2) | Average with Error Bar |
> |---------|-------------|-------------|------------------------|---------------------|---------------------|------------------------|
> | GSM8K   | Llama2-7B   | Llama2-13B  | 30.1%                  | 30.0%               | 29.9%               | 30.1% ±0.058%          |
> | GSM8K   | Gemma-2B    | Gemma-7B    | 44.8%                  | 44.6%               | 45.1%               | 44.8% ±0.15%           |
> | MBPP    | Llama2-7B   | Llama2-13B  | 40.2%                  | 40.3%               | 40.0%               | 40.2% ±0.088%          |
> | MBPP    | Gemma-2B    | Gemma-7B    | 51.2%                  | 50.9%               | 51.2%               | 51.1% ±0.10%          |
>
> **W4: [Domain]** We provide additional experimental results for legal and medical domains below. We note that MBPP already represents a specialized code domain task, and our results on GSM8K (mathematics) and BBH (challenging reasoning) further demonstrate effectiveness across diverse specialized capabilities beyond general knowledge.
>
>
> | Dataset | Source LoRA | Target      | Target no LoRA | Our Target LoRA | Acc Increase |
> |---------|-------------|-------------|----------------|-----------------|-------------|
> | Medical [1] | Llama2-7B   | Llama2-13B  | 33.8%          | 35.8%           | 2.0%        |
> | Medical [1] | Gemma-2B    | Gemma-7B    | 43.0%          | 45.1%           | 2.1%        |
> | Legal [2]  | Llama2-7B   | Llama2-13B  | 25.2%          | 40.8%           | 15.6%       |
> | Legal [2]  | Gemma-2B    | Gemma-7B    | 44.4%          | 48.4%           | 4.0%        |
>
> [1] Pal et al. MedMCQA: A Large-scale Multi-Subject Multi-Choice Dataset for Medical domain Question Answering
>
> [2] Chalkidis et al., LexGLUE: A Benchmark Dataset for Legal Language Understanding in English

---

> > ### Author Response · Authors · 2025-12-02
> >
> > **Q1: [Specialized Knowledge]** We define specialized knowledge broadly to encompass any systematic differences between base and fine-tuned models, including factual content, reasoning skills, or other capabilities acquired during fine-tuning. The specific nature of this knowledge is inherently data-dependent. Existing work has demonstrated that knowledge distillation via NLL loss successfully transfers both factual knowledge and reasoning abilities. Our use of perplexity difference maintains this generality by capturing the distributional shift between models, a signal that can correspond to diverse aspects of model adaptation regardless of the specific domain or capability being transferred.
> >
> > **Q2: [Prompt Diversity]** We demonstrate prompt diversity qualitatively via t-SNE plots in Figure 2. To quantify this further, we measured the type-token ratio (TTR) on the MBPP dataset. Sampling 100 prompts generated by our method versus Trans-LoRA's approach, we observe TTR values of 0.081 and 0.012, respectively. This confirms that Trans-LoRA prompts exhibit severely limited diversity, while our choice of prompt generator generates substantially more varied examples.
> >
> > **Q3: [Logit-based KD]** We kept our knowledge distillation component simple and consistent with established practices by performing supervised fine-tuning on the generated synthetic data, which typically uses NLL directly. While logit-based or KL-divergence-based distillation could be applied to the knowledge distillation stage, this choice is orthogonal to our core contribution: the data generation and filtering mechanism. Our focus is on identifying and synthesizing effective training examples, not on the specific distillation objective used.

---

### Meta-Review · Area_Chair_GNYV · 2026-01-10

**Summary:**

Despite the clear motivation and the authors' efforts during the rebuttal period, the consensus among reviewers is that the paper does not meet the bar for acceptance. The decision is primarily driven by 1) Limited Technical Novelty: A primary concern raised by multiple reviewers is that the core technical contribution—using perplexity is a well-established technique dating back significantly in the literature. 2) Ambiguity in Performance Drivers: The ablation studies did not sufficiently disentangle these factors to prove the efficacy of the filtering method over a strong generator baseline. 3) Experimental Scope: concerns remain regarding the scale of the experiments. The results are limited to relatively small models ($\le$ 13B), and it remains an open question whether this specific filtering method scales effectively to modern, large-scale architectures .

**Reviewer Concerns:**

The reviewers' concerns are mostly addressed, however there are some other concerns such as model novelty or on larger model are not addressed in this rebuttal.

**Reviewer Scores:**

NA

---

### Decision · Program_Chairs · 2026-01-26

Reject